# Dynamic co-catalysis of Au single atoms and nanoporous Au for methane pyrolysis

Wei Xi[1,3], Kai Wang [1,3], Yongli Shen[1,3], Mengke Ge[1], Ziliang Deng [1], Yunfeng Zhao[1], Qiue Cao[2], Yi Ding [1✉], Guangzhi Hu [2✉] & Jun Luo [1✉]

Nanocatalysts and single-atom catalysts are both vital for heterogeneous catalysis. They are recognized as two different categories of catalysts. Nevertheless, recent theoretical works have indicated that Au nanoparticles/clusters release Au single atoms in CO oxidation, and they co-catalyze the oxidation. However, to date, neither experimental evidence for the co-catalysis nor direct observations on any heterogeneous catalysis process of single-atom catalysts are reported. Here, the dynamic process of nanoporous Au to catalyze methane pyrolysis is monitored by in situ transmission electron microscopy with high spatial–temporal resolutions. It demonstrates that nanoporous Au surfaces partially disintegrate, releasing Au single atoms. As demonstrated by DFT calculation, the single atoms could co-catalyze the reaction with nanoporous Au. Moreover, the single atoms dynamically aggregate into nanoparticles, which re-disintegrate back to single atoms. This work manifests that under certain conditions, the heterogeneous catalysis processes of nanocatalysts and single-atom catalysts are not independent, where their dynamic co-catalysis exists.

[1] Center for Electron Microscopy and Tianjin Key Lab of Advanced Functional Porous Materials, Institute for New Energy Materials & Low-Carbon Technologies, School of Materials Science and Engineering, Tianjin University of Technology, 300384 Tianjin, China. [2] School of Chemical Science and Technology, Key Laboratory of Medicinal Chemistry for Nature Resource, Ministry of Education, Yunnan University, 650091 Kunming, China. [3]These authors contributed equally: Wei Xi, Kai Wang, Yongli Shen. ✉email: yding@tjut.edu.cn; guangzhihu@ynu.edu.cn; jluo@tjut.edu.cn

eterogeneous catalysis is a crucial process in industries of chemicals, energy and environmental protection and their related fundamental researches, such as hydrogenation[1–3], CO oxidation[4–8], and CH$_4$ activation and use[9–16]. In its currently increasing development, nanocatalysts[1–3,7,9–20] and single-atom catalysts (SACs)[4,8,21–28] are two different categories of vital heterogeneous catalysts. The two categories have exhibited excellent performances and complementary functions, and their catalytic processes/mechanisms are generally considered to be different from each other[24,25,28]. Nevertheless, two recent works[5,6] use theoretical calculations to indicate that during the CO oxidation catalyzed by Au nanoparticles/clusters, Au single atoms (SAs) can break away from the nanoparticles/clusters, and the SAs and the nanoparticles/clusters co-catalyze the CO oxidation. These results give rise to an inspiration that the gap between the heterogeneous catalytic processes of nanocatalysts and SACs would be bridged by the co-catalysis. Significantly, this inspiration is indirectly supported by experimental observations that metal nanoclusters or substrates can release metal SAs and they co-catalyze the growth of carbon nanostructures in all-solid-phase catalysis[29–31]. However, up to now, no experimental evidence has been reported for the emergence of SACs in any heterogeneous catalysis process of nanocatalysts. Meanwhile, even though the dynamic processes of metal SAs to catalyze the growth of carbon nanostructures in all-solid-phase catalysis have been impressively demonstrated[27,29–31], no published works in the literature give direct observations on any heterogeneous catalysis process of SACs. Thus, it remains elusive whether the co-catalysis exists in any heterogeneous catalysis process initiated by either of nanocatalysts and SACs.

Among heterogeneous catalysis reactions, the methane (CH$_4$) pyrolysis is a fully green single-step technology for producing nanocarbon and hydrogen (H$_2$), the most environmentally friendly energy carrier[14–16]. It is also important for mitigating the environmental challenge associated with CH$_4$ emission[12,13,15,16]. Meanwhile, Au nanocatalysts and their derived catalysts have exhibited excellent catalytic activities for CH$_4$ activation and use[9,10,14]. As a new type of Au nanocatalyst with three-dimensional (3D) bicontinuous porous structure[32–38],

nanoporous gold (NPG) has shown high performances in heterogeneous catalysis[32,36,37]. Its unique structure has large specific surface area, low density, and high permeability, which are all advantageous for heterogeneously catalytic reactions[32–38]. Therefore, we here use NPG to catalyze the CH$_4$ pyrolysis and characterize its dynamic catalysis process by conventional and in situ transmission electron microscopy (TEM) with high spatial and temporal resolutions. The characterizations reveal that some of the NPG surfaces continuously disintegrate, from which Au SAs are released. The Au SAs and NPG co-catalyze the CH$_4$ pyrolysis to produce amorphous carbon and H$_2$. Moreover, during the catalysis process of the Au SAs, some SAs dynamically aggregate into Au nanoparticles, and the nanoparticles dynamically re-disintegrate back to SAs. The nanoparticles and SAs also co-catalyze the CH$_4$ pyrolysis, which is confirmed by our theoretical calculations. This work gives solid experimental evidence that under certain conditions of heterogeneous catalysis, SAs are released by nanocatalysts and then participate in catalytic reactions originally initiated by nanocatalysts, and vice versa. These results can help to bridge the gaps between the catalysis processes of nanocatalysts and SACs and provide new inspirations for designing the catalysts.

## Results

**Characterization of NPG before and after ex situ CH$_4$ pyrolysis.** Ex situ catalytic CH$_4$ pyrolysis reaction was performed using NPG as the catalyst at 580 °C in a system of direct-current plasma-enhanced chemical vapor deposition (dc-PECVD). The flow rates of CH$_4$, Ar, and H$_2$ were 50, 200, and 20 sccm (sccm is standard cubic centimeters per minute), respectively, and their chamber pressure was 5 mbar (see "Methods" for more details; it should be noted that this dc-PECVD system is independent from and outside TEM). The structures of NPG before and after the ex situ reaction were investigated firstly by conventional scanning electron microscopy (SEM) and TEM techniques including high-angle annular dark field (HAADF) imaging of scanning TEM (STEM), as shown in Fig. 1 and Supplementary Figs. 1–4.

The NPG samples were prepared by dealloying, and they possess bicontinuous pore channels and ligaments[32–38]. For

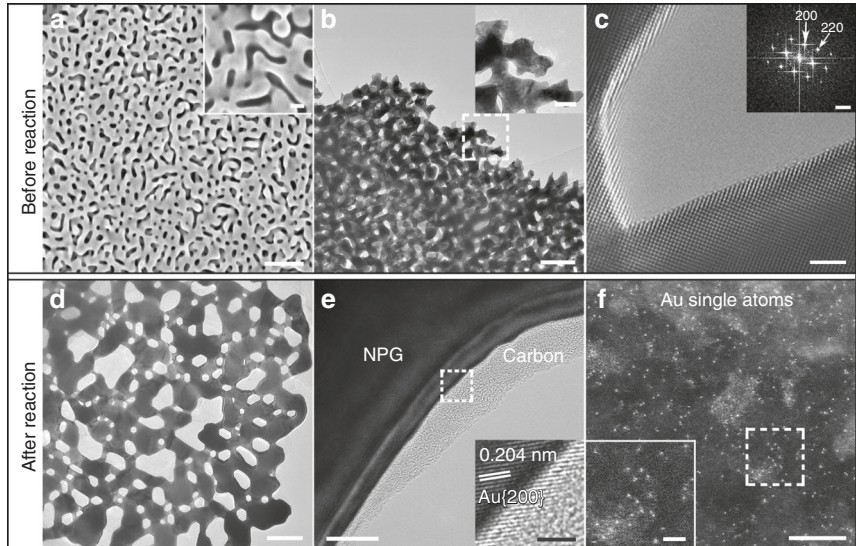

**Fig. 1 Characterization of NPG before and after the ex situ catalytic CH$_4$ pyrolysis reaction.** SEM (**a**) and TEM (**b**) images of an NPG sample before the reaction. The insets in **a** and **b** are a high-magnification SEM image and the close-up view of the boxed region in **b**, respectively. **c** HRTEM image and its fast Fourier transform pattern. TEM (**d**) and HRTEM (**e**) images of an NPG sample after the reaction. **f** HAADF image of a carbon region on the NPG sample. The insets in **e** and **f** are the close-up views of the boxed regions in **e** and **f**, respectively. Scale bars: **a** 200 nm; the inset of **a** 50 nm; **b** 200 nm; the inset of **b** 50 nm; **c** 2 nm; the inset of **c** 5 nm$^{-1}$; **d** 200 nm; **e** 10 nm; the inset of **e** 2 nm; **f** 5 nm; the inset of **f** 2 nm.

instance, the as-prepared sample in Fig. 1a–c has the average ligament size of ~50 nm. Its energy-dispersive X-ray spectroscopy result is displayed in Supplementary Figs. 1 and 2, indicating that it is Au containing $1.37 \pm 0.38$ at% Ag (0.38 is the standard deviation, and the Source data are provided as a Source Data file) and the Ag distribution in it is homogeneous. The high-resolution TEM (HRTEM) image in Fig. 1c shows that the NPG sample has the standard crystal structure of Au, and the surfaces of its ligaments are clean. After the reaction, most of the pores become larger, some ligaments slim and the others coarsen, amorphous carbon has been produced on the surfaces of all ligaments, and the ligaments are still Au (Fig. 1d, e). Importantly, HAADF imaging shows that a high density of heavy metal SAs appear in the amorphous carbon (Fig. 1f, Supplementary Figs. 3–8, 14 and Supplementary Notes 1–4, 6) considering C is substantially lighter than Au or Ag[4,8,21–26,28,39–41]. While no Ag signals were detected within the amorphous carbon (Supplementary Figs. 4 and 11), possibly due to its very low content in the original NPG sample, the existence of Ag SAs is possible. Contrast intensity analysis in Supplementary Figs. 5–8 and Supplementary Notes 1–4 show the overwhelming majority of bright spots are Au SAs, and a few darker bright spots may be Ag SAs (Fig. 1f, Supplementary Figs. 3–6, 8, 14 and Supplementary Notes 1, 2, 4, 6), especially considering the contrast intensity is also related to its distance from the focal plane (Supplementary Fig. 6 and Supplementary Note 2).

**Dynamic process of the $CH_4$ pyrolysis reaction on NPG.** According to the results in Fig. 1, it is speculated that during the above-described $CH_4$ pyrolysis reaction, the NPG surfaces experienced an evolution, causing the release of Au SAs from the surfaces. To check this speculation, in situ HRTEM was used to monitor the dynamic process of the $CH_4$ pyrolysis reaction on NPG at 346 °C (see details in "Methods"). It should be noted that TEM and STEM images of NPG are all projections of its 3D structure, in which two types of surface regions with positive and

negative curvatures exist[34,35]. The dynamic processes on the two types were both recorded, as displayed in Fig. 2 and Supplementary Movies 1 and 2, which have been slowed down by eight times of the real rate (80 frames per second) for facilitating visual inspection. This rate means a high temporal resolution of 0.0125 s, and Fig. 2 indicates that the spatial resolution of the in situ HRTEM technique is high enough to resolve the Au crystal lattices of NPG. More importantly, the in situ observations show that on both the two types of surface regions, the thicknesses of the amorphous carbon increased continuously, suggesting that the catalytic $CH_4$ pyrolysis reaction continuously occurred on the surfaces. At the same time, the diameters of the two NPG ligaments decreased continuously, indicating that their surfaces continuously disintegrated, while the crystal structures within the residual ligaments were still Au because the HRTEM images show that their crystal lattices are the same as those of standard Au.

The diameter decrease of the slimmed NPG ligaments indicates that the extensive emergence of the Au SAs (Fig. 1f) was due to that a large number of Au atoms were released from the surfaces of the slimmed ligaments and were then dispersed as SAs in the amorphous carbon layers. Meantime, the diameter increase of the coarsened ligaments manifests that some of the Au SAs came back to some ligaments. These findings are very similar to the theoretically reported results that the adsorption and diffusion of CO molecules in the form of Au–CO species on a $CeO_2$ substrate cause Au SAs to break away from Au nanoparticles/clusters and the Au SAs can come back to the nanoparticles/clusters[5,6]. Thus, we performed density functional theory (DFT) calculations, and the result shows that C atoms in the amorphous carbon can adsorb on the surface of an Au crystal, and their interaction with the crystal surface pulls Au atoms out of the surface, leading to the formation of Au SAs (Supplementary Fig. 9).

It is well known that carbon deposits generally prevent the contact between reactants and catalyst surfaces and thus stop heterogeneous catalytic reactions[13,15,16]. But, the $CH_4$ pyrolysis reaction on the surfaces of NPG still proceeded normally even

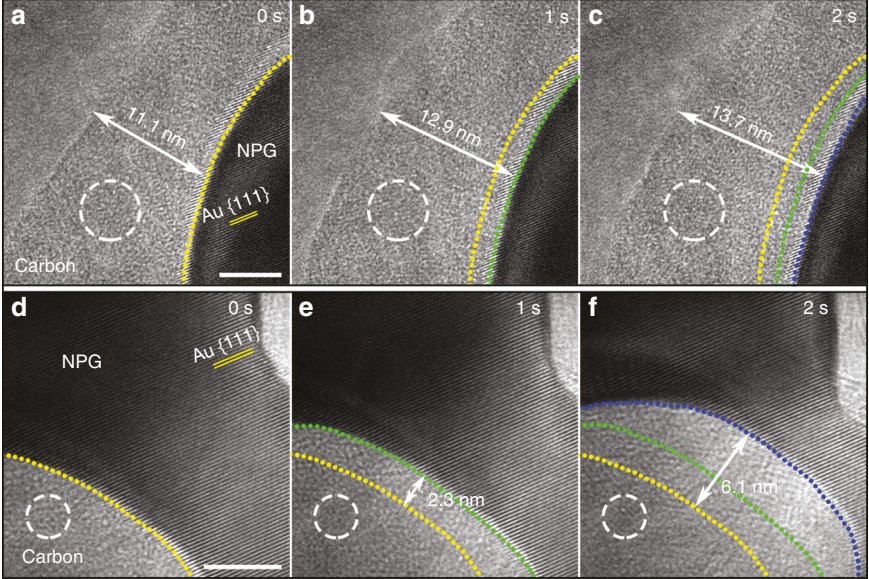

**Fig. 2 In situ dynamic process of the catalytic $CH_4$ pyrolysis reaction on NPG. a–c** HRTEM images at three different moments during the reaction on a surface region with positive curvature. The images have the same scale bar. The moment in **a** is defined as 0 s. The yellow, the green, and the blue dashed lines indicate the positions of the surface at 0, 1, and 2 s, respectively. The white dashed circles denote the positions of a randomly selected carbon region, and the arrows indicate the thickness of the amorphous carbon at 0, 1, and 2 s. **d–f** have the same meanings as **a–c** but are for a surface region with negative curvature. The arrows indicate the reduction of the NPG ligament at 0, 1, and 2 s. **a–c** and **d–f** are cut from Supplementary Movies 1 and 2, respectively (see more details in "Methods"). Scale bars: **a** 5 nm; **d** 5 nm.

when the thicknesses of the amorphous carbon layers exceeded 10 nm (Fig. 2 and Supplementary Movies 1 and 2). This phenomenon is due to the presence of abundant nanopores with diameters of 3–7 nm and sub-nanopores with diameters centered at 0.82 nm in the amorphous carbon layers (Fig. 3a and Supplementary Fig. 10), which serve as channels for the gas exchange through the amorphous carbon layers and thus effectively alleviate the poisoning effect that normally occurs on the catalyst surface. These pores should be produced by the flows of the gas-phase products ($CH_3$, $CH_2$, CH, and $H_2$) during the $CH_4$ pyrolysis, which is similar to the pore generation in gas–eutectic transformations[42].

The presence of the gas channels also allows the Au SAs in the amorphous carbon layers to contact $CH_4$ molecules and thus catalyze their pyrolysis, which is proven by the HRTEM images in Figs. 2 and 3 and Supplementary Movies 1–4. First, the images in Fig. 2 and their related movies (Supplementary Movies 1 and 2) show that the contrasts within the amorphous carbon layers including those carbon parts distant from the NPG surfaces were always dramatically changing, when the conditions of the $CH_4$ pyrolysis reaction were kept. For instance, the carbon zone indicated by the dashed circles in Fig. 2a–c was distant from the NPG surface, but it had a changing contrast during the short period of 2 s. So did the one in Fig. 2d–f. For visual inspection, the contrast changing in the corresponding movies is clearer. In contrast, after the $CH_4$ flow was stopped, the contrast changing became from violent to rather mild and even vanished, as shown by Fig. 3b, c and Supplementary Movie 3. For instance, the contrasts of the six zones indicated by the dashed circles in Fig. 3b, c were stable and almost unchanged. The above

comparison manifests that the violent changing of the contrast within the amorphous carbon parts distant from the NPG surfaces in Fig. 2 was not due to the electron beam of TEM but to a certain reaction in the carbon parts.

Further, Fig. 3d–f and its related movie (Supplementary Movie 4) show that in an amorphous carbon region distant from the NPG surface, a nanoparticle with the Au crystal structure temporarily formed and then quickly disappeared before 0.4125 s during the $CH_4$ pyrolysis. Meantime, the contrast within the carbon region was also changing, such as the contrast of the zone indicated by the dashed circles in Fig. 3d–f (see Supplementary Movie 4 for clearer visual inspection). These phenomena were easily found in other amorphous carbon regions (Supplementary Figs. 11, 12 and Supplementary Movie 5). Due to the imaging mechanism of HRTEM that is different from that of HAADF[43], Au SAs cannot be imaged by the HRTEM images in Figs. 2 and 3 and their related movies. Nevertheless, if no sources of Au atoms had existed in the amorphous carbon layers, the Au nanoparticles would not have formed. Moreover, we have randomly selected 20 different regions of amorphous carbon for HAADF imaging and found that all of them contain Au SAs. Besides, as analyzed before, the changing of the contrasts within the amorphous carbon regions suggest a certain reaction occurring in the carbon parts. This reaction has to be the $CH_4$ pyrolysis catalyzed by Au SAs, because no other reactions can exist. Therefore, the above analyses imply the following. The Au SAs and NPG co-catalyzed the $CH_4$ pyrolysis reaction. At the same time, in amorphous carbon regions distant from NPG surfaces, some Au SAs dynamically aggregated into Au nanoparticles (Fig. 3d–e, Supplementary Figs. 11–14, Supplementary Notes 5, 6 and

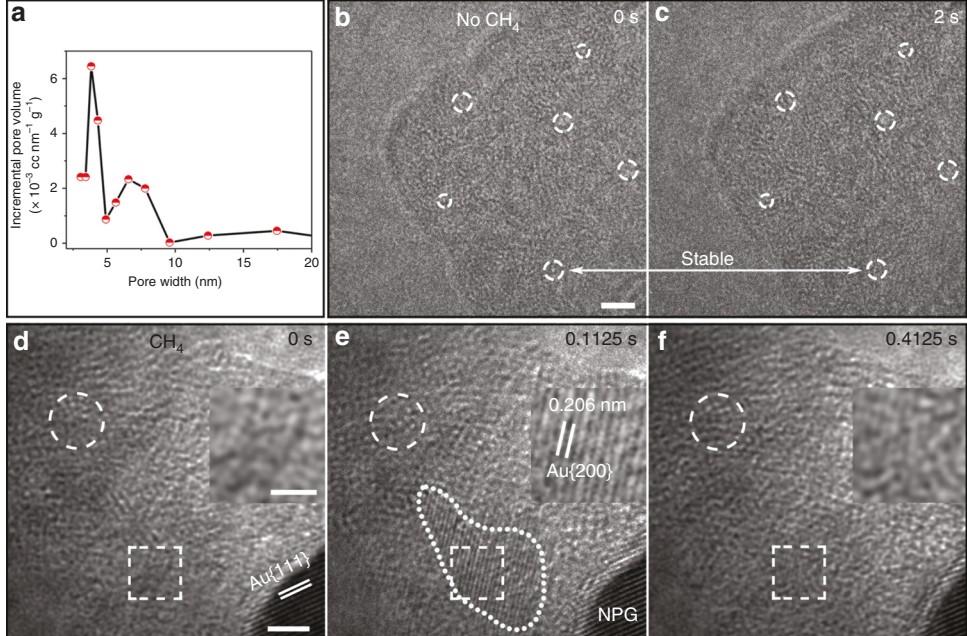

**Fig. 3 Characterization of the amorphous carbon layers and in situ dynamic catalysis process of Au SAs. a** Size distribution of nanopores in the amorphous carbon layers produced by the ex situ $CH_4$ pyrolysis reaction (see more details in "Methods" and Supplementary Fig. 10). **b, c** HRTEM images of an amorphous carbon layer at two different moments after the layer was produced and the $CH_4$ flow was stopped with the other conditions of the in situ $CH_4$ pyrolysis reaction unchanged. The two images have the same scale bar. **d–f** HRTEM images of an amorphous carbon layer at three different moments during the reaction with all the conditions unchanged. The moments in **b** and **d** are defined as 0 s. The white dashed circles in **b** and **c** denote the positions of six randomly selected carbon zones at 0 and 2 s, which are typical unchanged regions, and the ones in **d–f** indicate the positions of a randomly selected carbon zone at 0, 0.1125, and 0.4125 s, which is a typical changed region. The boxes in **d–f** indicate the positions of a nanoparticle that temporarily formed and disappeared. The insets in **d–f** are the close-up views of the boxed regions in **d–f**, respectively. The dashed closed curve in **e** indicates the boundary of a temporarily formed crystal nanoparticle. **b, c** and **d–f** are cut from Supplementary Movies 3 and 4, respectively (see more details in Methods). Scale bars: **b** 2 nm; **d** 2 nm; the inset of **d** 1 nm.

Supplementary Movies 4, 5), which re-disintegrated back to SAs (Fig. 3e–f, Supplementary Figs. 12, 13, Supplementary Note 5 and Supplementary Movies 4–6), similar to the catalysis-induced surface disintegration and ligament coarsening in Figs. 1d and 2. This phenomenon suggests that the Au SAs and the nanoparticles also co-catalyzed the $CH_4$ pyrolysis reaction, which is confirmed by DFT calculations below.

## Discussion

The catalytic reactivity of Au SAs toward the $CH_4$ pyrolysis was checked by performing DFT calculations on a model of an Au SA anchored on an amorphous carbon substrate. The DFT result is shown in Supplementary Fig. 15, indicating that a $CH_4$ molecule is firstly adsorbed on a C atom of the substrate by a weak interaction. Then, the four H atoms are gradually removed to form the final product C through the interactions between the intermediates, the Au SA and the substrate. The DFT result also manifests that the above process is feasible under the actual reaction conditions.

The experimental aggregation of Au SAs into Au nanoparticles during the catalytic $CH_4$ pyrolysis is easily understandable because Au SAs have much higher surface energies than their counterpart nanoparticles/clusters[23–25,28]. To gain insight into the re-disintegration of the temporarily formed Au nanoparticles, we have also performed DFT calculations. Figure 4 displays the calculated results. Initially, an Au cluster is loaded on amorphous carbon (Fig. 4a). Figure 4b, c shows that when the intermediate $CH_3^*$ of the $CH_4$ pyrolysis is adsorbed on the Au cluster, the distances between the Au atom with $CH_3^*$ adsorbed and the neighbors in the cluster become from 3.42 and 2.78 Å to 4.52 and 4.67 Å. That is, the adsorption of $CH_3^*$ weakens the interaction between the Au atom and the cluster, and thus this Au atom is inclined to break away from the cluster. Similar results are also found in $CH_2$–Au/C, CH–Au/C, and C–Au/C (Supplementary Fig. 16). The energy change from an initial state (IS) to its corresponding transition state (TS) is the activation energy ($E_a$). Figure 4d gives the $E_a$ values of the configuration evolutions of $CH_3$–Au/C, $CH_2$–Au/C, CH–Au/C and C–Au/C to be 0.77, 0.01, 0.31, and 0.47 eV, respectively. These values are all easily achievable under the actual reaction conditions, such as the high temperatures of 580 and 346 °C[44]. That is, all of the final states (FSs) in Fig. 4b–d can be easily realized under the actual reaction conditions, indicating not only the efficient catalytic reactivity of

the Au cluster toward the $CH_4$ pyrolysis but also the feasibility of the re-disintegration of the temporarily formed Au nanoparticles into Au SAs. Moreover, we have also calculated Au clusters composed of 3 and 2 atoms (namely $Au_3$ and $Au_2$) and found that before the clusters can catalyze the conversion of $CH_3^*$ into $CH_2^*$, they should have broken into three or two Au SAs (see more details in Supplementary Figs. 17 and 18).

In addition to the dynamic processes of Au SAs and NPG ligaments we observed and calculated above, in situ TEM studies of CO oxidation on NPG[34,38] revealed dramatically enhanced surface diffusion of Au which led to substantial coarsening of some pores and some Au ligaments. In our work, the coarsening also occurs during the $CH_4$ pyrolysis, as indicated by Fig. 1b, d. Our in situ observation shows that the breaking and migration of a ligament can make its adjacent ligament coarsen (see details in Supplementary Fig. 19).

It should be noted that the incident high-energy (200 keV) electrons might influence the in situ TEM experiments. To explore this, we performed continuous electron-beam irradiation experiments on NPG at 23 °C and during heating without and with $CH_4$, and an on-line mass spectrometer was used to detect hydrogen production when the electron beam was turned off (see details in Methods and Supplementary Figs. 19–25, Supplementary Note 7). The results clearly indicate that not the electron beam but the NPG-catalyzed $CH_4$ pyrolysis plays the dominant role in the NPG structure evolution and the migration of Au atoms from the NPG surfaces (see more details in Supplementary Figs. 23–25 and Supplementary Note 7). This conforms to a published theoretical result[45], which indicates that the minimum incident-electron energy to knock out Au atoms is approximately 407 keV, much higher than 200 keV of the electron irradiation used in our work.

It should be noted that the recorded hydrogen production profiles on NPG samples with different Ag contents indicate that the residual Ag may contribute to the catalytic $CH_4$ pyrolysis, especially at temperatures below 200 °C (see more details in Supplementary Figs. 11, 26 and Supplementary Note 8). In order to evaluate the role of the residual Ag of NPG in the $CH_4$ pyrolysis, we fabricated a series of NPG samples with different Ag contents and detected their hydrogen production (see details in Methods and Supplementary Fig. 26, Supplementary Note 8). The results show that with the decrease of the Ag content, the low-temperature (<200 °C) activity decreases remarkably, and for our discussed NPG sample with the lowest Ag content of 1.37 ± 0.38

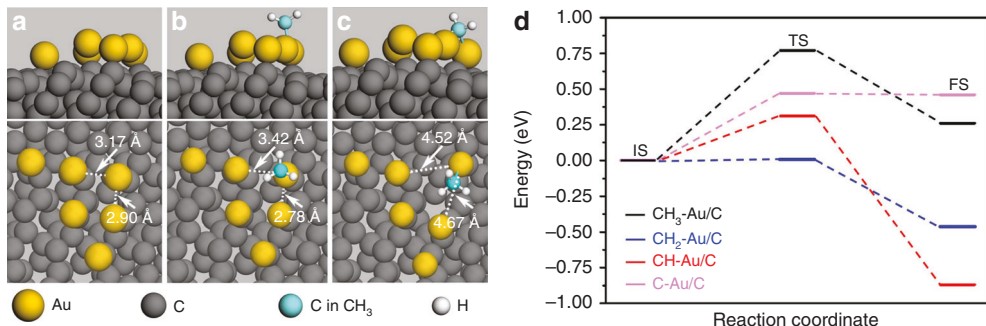

**Fig. 4 DFT-calculated configuration evolutions of an Au cluster on an amorphous carbon. a** DFT-relaxed atomic configuration of a pure Au cluster consisting of six Au atoms on amorphous carbon, which is denoted as Au/C. The top and the bottom panels are the side and the top views of the configuration, respectively. **b, c** DFT-relaxed atomic configurations of the IS (**b**) and the FS (**c**) of $CH_3$–Au/C (see Supplementary Fig. 16 for the configurations of $CH_2$–Au/C, CH–Au/C and C–Au/C). $CH_3$–Au/C, $CH_2$–Au/C, CH–Au/C and C–Au/C denote the configurations of the Au cluster, respectively, with the presence of $CH_3^*$, $CH_2^*$, $CH^*$ and $C^*$, which are the intermediates and the carbon product of the $CH_4$ pyrolysis. IS, TS and FS are the initial, the transition and the final states, respectively. The dashed lines pointed by the white arrows in **a–c** denote the interatomic distances. **d** Activation energies of the configuration evolutions of the Au cluster, in which the black, blue, red, and pink lines represent the corresponding IS, TS, FS and activation energy of the Au cluster evolution under the adsorption of $CH_3$, $CH_2$, CH, and C, respectively.

at%, it becomes active only when the reaction temperature reaches 200 °C, distinct from other samples with higher Ag contents. Considering the catalytic activity of NPG itself has a significant structural dependence[34–36,38], we cannot simply infer from the Supplementary Fig. 26 and Supplementary Note 8 whether there is a linear relationship between the Ag content and its catalytic activity for methane pyrolysis.

Actually, we selected this particular sample (with such a low Ag content) because previous report[14] and our experiment (see details in Methods and Supplementary Fig. 27) prove that $CH_4$ can be pyrolyzed by the catalysis of pure gold, and the drastic structural change of this particular sample (with such a low Ag content) provides clear evidence of how Au ligaments restructure and how Au SAs form, aggregate and re-disintegrate (Figs. 2 and 3 and Supplementary Movies 1, 2, 4 and 5). It is quite unlikely for such low Ag content to cause such drastic change processes, and no Ag signals were detected in the carbon layers (Supplementary Figs. 4 and 11). On the other hand, a series of transition metal clusters have been studied[46,47] for their abilities to catalyze the pyrolysis of methane, and the results show that the overall ability of Ag to activate C–H bonds of methane is actually slightly lower than that of Au (Ag is only slightly better than Au in activating the fourth C–H bond of methane)[48] (Supplementary Table 1). Considering the possible differences in actual structures of the catalysts (Au, Ag, AuAg alloys, nanoparticles, or NPG with different size), it is thus reasonable to propose that the ability of Ag to catalyze the methane pyrolysis would not be substantially stronger than that of Au. This is the reason to speculate that in the case of extremely low Ag content, the drastic dynamic catalytic behavior of the NPG catalyst should be dominated by Au.

In summary, NPG is discovered to release a large number of Au SAs during its catalysis toward the methane pyrolysis. The dynamic process of this heterogeneous catalysis is directly observed by in situ HRTEM with high spatial and temporal resolutions, indicating that the NPG surfaces partially disintegrate. This disintegration is continuous, by which Au SAs are produced. As demonstrated by DFT calculation, the Au SAs can co-catalyze the reaction with nanoporous Au. At the same time, some of the SAs dynamically aggregate into Au nanoparticles, which temporarily exist and dynamically re-disintegrate into Au SAs. The Au SAs and the nanoparticles also co-catalyze the methane pyrolysis. These findings are confirmed by DFT calculations. This work manifests that the heterogeneous catalysis processes of nanocatalysts and SACs may need to be investigated integratedly, and the designs of nanocatalysts and SACs may not be independent from each other.

## Methods

**Sample preparations**. The NPG samples were synthesized by dealloying 12-karat Au–Ag alloy films with a thickness of 100 nm in concentrated $HNO_3$ at 30 °C for 30 min. Then, the dealloyed films were immersed in ultrapure water for several times to clean them. Finally, the films floated on the surface of ultrapure water, and they were NPG pieces. This synthesis is the same as reported[34–38]. For conventional and in situ TEM characterizations, conventional TEM grids and sample chips from the DENS solutions company at The Netherlands were used, respectively, to collect NPG pieces from the water surface. In particular, for the in situ study, no plasma was used to treat NPG pieces before they were inserted into TEM. For other characterizations and the ex situ $CH_4$ pyrolysis, some NPG pieces were dried under an infrared lamp. In addition, we used a magnetron sputtering instrument (JGP-450C, SKY Technology Development Co., Ltd, Chinese Academy of Sciences) with a pure Au (99.99%) and $Au_1Ag_1$ (99.99%) target to produce pure Au nanoparticles as a control sample and Au and Ag SAs co-existing on ultra-thin carbon films (Beijing XXBR Technology Co., Ltd).

**Ex situ $CH_4$ pyrolysis**. The pyrolysis was performed in a dc-PECVD system (Black Magic 6-inch, AIXTRON). An NPG sample was loaded into the reaction chamber and was heated to 580 °C under 200 sccm Ar and 20 sccm $H_2$ with a chamber pressure of 5 mbar. Then, the plasma (80 W power) was turned on, and 50 sccm $CH_4$ was introduced into the chamber for 1–2 h. Afterwards, the plasma and the

$CH_4$ flow were stopped. This experiment can rapidly produce a mass of carbon-loaded NPG with SAs, whose structures are shown in Fig. 1d, f and quite similar to those of the samples through the in situ study (Supplementary Figs. 28 and 29).

**Pore size distribution**. After the ex situ $CH_4$ pyrolysis, 100 mg NPG with amorphous carbon was used to evaluate the pore structure of the amorphous carbon, which was performed using $N_2$ (77 K) and $CO_2$ (273 K) adsorption–desorption measurements on an Autosorb-IQ-MP Micromeritics analyzer. Pore volumes were computed by the Barrett–Joyner–Halenda (BJH) method, the size distribution of nanopores was determined by the Brunauer–Emmett–Teller (BET) method by using the $N_2$ adsorption branch under 77 K, and the size distribution of sub-nanopores was determined by the nonlocal density functional theory model based on the $CO_2$ adsorption branch at 273 K.

**TEM experiments**. The conventional TEM characterizations were performed with three TEM instruments of Talos F200X, Titan Cubed Themis G2 300 and Themis Z from FEI at 200 kV. For each of the in situ TEM experiments, a DENSsolutions chip with NPG pieces was assembled into an in situ TEM gas-phase holder (Climate S3 +, DENSsolutions), whose specified range of error for temperature is <5%. The holder was then inserted into Talos F200X with a high-speed camera (Ceta 2, FEI) to record in situ TEM movies/images at 200 kV, and the dose rate of the electron beam applied for the in situ experiments was 0.039–1570 eÅ$^{-2}$ s$^{-1}$, which was adjusted by the magnification (×4300–×1,050,000). Before the in situ catalytic reaction, the gas line of the holder was flushed with Ar for 30 min at atmospheric pressure, and $CH_4$ was then introduced. Afterwards, the temperature was increased by 30 °C min$^{-1}$ to 346 °C, at which we found amorphous carbon to start to grow significantly and then recorded movies with the HRTEM mode and the camera. The gases of pure $CH_4$ (99.995%) and Ar (99.999%) were purchased from Beijing ZG Special Gases Science & Technology Co., Ltd (Beijing, China). The on-line mass spectrometer used to detect hydrogen production was Omnistar GSD 320 O1 (Pfeiffer Vacuum, Germany). For facilitating visual inspection, Supplementary Movies 1–5 have been slowed down by eight times of the real rate (80 frames per second), in which the moments corresponding to the images in Figs. 2 and 3b–f and Supplementary Fig. 12 are paused for a short while. The HRTEM images in Figs. 2 and 3d–f and Supplementary Fig. 12 are Wiener-filtered, and the Wiener filtering has been widely used to remove noise in HRTEM and high-resolution STEM images[27,49].

**DFT calculations**. The calculations were performed using the generalized gradient approximation-Perdew, Burke, and Ernzerh of (PBE) functional[50] as implemented in the all-electron DMol3 code[51,52]. The double numerical plus polarization basis set was used throughout the calculations. The convergence criteria were set to be $2 × 10^{-5}$ Ha, 0.004 Ha Å$^{-1}$ and 0.005 Å for the energy, the force and the displacement convergences, respectively. A self-consistent field density convergence with a threshold value of $1 × 10^{-5}$ Ha was specified. In the computation of the formation of Au SAs from NPG surfaces, 3 × 3 Au {311} cells with the thickness of six atomic layers were used to represent the NPG surface. In the computation of the methane pyrolysis reaction mechanism, an amorphous carbon layer of 8.65 Å × 9.98 Å × 6 Å was selected as the carbon carrier, on which an Au SA was located. In the computation of the configuration evolutions of the Au cluster, an amorphous carbon layer of 14.5 Å × 15.0 Å × 4 Å was selected to represent the amorphous carbon carrier, on which six Au atoms were placed to simulate the Au cluster. In all cases, a vacuum region of 30 Å was used to ensure negligible interaction between periodic replicas. All the TSs were determined by the linear synchronous transit and quadratic synchronous transit methods. The TS structures were characterized by analyzing the vibrational normal modes and by using a local minimum search (after a small distortion of each TS in the reaction coordinate direction) to reach the reactants and products[53].

## Data availability

The data that support the findings of this study are available within the article (and its Supplementary Information files) and from the corresponding authors upon reasonable request. The source data underlying the Ag content of 1.37 ± 0.38 at% are provided as a Source Data file.

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

## Acknowledgements

This work was financially supported by National Key R&D Program of China (2017YFA0700104), National Science Fund for Distinguished Young Scholars (51825102), National Natural Science Foundation of China (51671145, 21506148, 51761165012, and 51971157), Tianjin Science Fund for Distinguished Young Scholars (19JCJQJC61800), and National Program for Thousand Young Talents of China. We acknowledge the National Supercomputing Center in Shenzhen for providing the computational resources and materials studio (version 7.0, DMol3). We thank Mr Zhaolong Chen at Peking University for assistance with the PECVD system and Professor Botao Qiao at Dalian Institute of Chemical Physics for useful discussion.

## Author contributions

W.X. and K.W. co-performed the TEM experiments and the CH₄ pyrolysis reactions. Y.S. carried out the theoretical calculations. M.G. prepared the NPG samples. Z.D. assisted in the TEM experiments. Y.Z. performed the gas adsorption–desorption experiments and analyzed the data, to which W.X., Q.C., G.H., and J.L. contributed. Y.D., G.H., and J.L. co-supervised the project. All authors contributed to the data analyses and discussion and wrote the manuscript.

## Competing interests

The authors declare no competing interests.
