## [Peer Review File · Nature Communications]

Reviewers' comments:

Reviewer #1 (Remarks to the Author):

In their manuscript „Dynamic co-catalysis of Au single atoms and nanoporous Au for methane pyrolysis“ the authors demonstrate the dynamics of methane pyrolysis catalyzed by nanoporous Au observed by in situ TEM. The disaggregation of Au catalyst during the heterogeneous catalytic reaction is observed, which is very important for the understanding of the behavior of nanocatalyst in reactions. They present a careful work with high technical standard and together with the main claim that the released Au single atoms and the nanoporous Au co-catalyze the pyrolysis, it would be appropriate to publish in Nature Communication. However before I can recommend the paper for publication in Nature Communication, I would like the authors to present more results to justify this finding, as there are:

1, The evidence of the existence of Au single atoms.

As described in the manuscript, the Au single atoms can be found in the carbon structures generated by methane pyrolysis. The only evidence of Au single atoms provided is the HAADF image of Fig. 1f. Considering the concept of Au single atom is important to this work. A low magnification HAADF image of an area showing the boundary of nanoporous Au and carbon structure (like Fig. 1e) with a high magnification HAADF image showing the Au single atom of an area from the carbon part are necessarily. In addition and EDX map would be very helpful to understand justify the Au distribution.

2, The disintegration of nanoporous Au.

By comparing the structures of nanoporous Au before and after reaction (Figs 1b and 1d), it is clear that with increasing pore size, the Au areas also increase. That means the nanoporous Au not only disintegrates but also restructures during the reactions. Could you please comment to this fact.

3, The role of Au single atoms during reactions.

The co-catalysis of Au single atoms and nanoporous Au is the key idea of this work. The Au single atoms could also catalyze the methane pyrolysis as suggested by the DFT calculation. What is the DFT result if you have little Au clusters maybe composed of 2-3 atoms? However, there is no corresponding experimental evidence for the effect. I understand that experimental evidence is rather tricky to proof, otherwise you must weaken your statement such as :as 'DFT calculation demonstrates that the released Au single atoms and the nanoporous Au could co-catalyze the pyrolysis...' in the abstract, introduction and conclusion parts.

4, The influence of electron beam.

The dose rate of the e-beam applied for the TEM experiments should be provided. The incident electrons with high energy (200 keV) could also influence the in situ experiments. The incident electrons could transfer kinetic energy to the Au nanocatalyst and even sputter the Au atoms on the surface. Thus I would like to request to discuss the possible influence of the electron beam in the manuscript.

5, The pore size distribution.

The samples are treated by a mixture of 15 ml HCl (37%) and 5 ml HNO₃ (65%) before the measurement of pore size distribution. Why you removed the nanoporous Au? And will the strong oxidizing mixture solution influence the structure of amorphous carbon? In addition, nitrogen adsorption-desorption measurement is only suitable for studying the nanopores of carbon structure. It might be advised to use CO₂ adsorption-desorption measurement to study the sub-nano pores of carbon structure.

6. Technical details

Fig. 1c is highly defocused. Please provide an in-focus image. In general, the Fourier transform pattern of a HRTEM image shows more clearly all the frequencies present in the image, then just to mark selected lattice fringe spacings in the image.

Reviewer #3 (Remarks to the Author):

The authors attempted to visualize the role of individual Au atoms played in the reaction with methane on the surface of a nanoporous Au (NPG) leaf specimen by applying in-situ transmission electron microscopy. The challenge to push in-situ observation forward is highly important and useful for elucidating the catalytic reactions in real space and time. Therefore, any study as this manuscript potentially deserves for publication. After reading this manuscript, it is however concluded that the experiments by the authors were not well organized and the analyses were vague even though the title of the manuscript appeared to be fascinating. The readership of Nature Communications will be misled by the manuscript.

In any NPG leaf specimen that is prepared by the dealloying method, residual silver inevitably remains over 1 at.%. Fujita et al. already reported in their paper (Nat. Mater. 11 (2012) 775.) that silver was included in a NPG leaf specimen by 1.2 at. % after the dealloying process even for a prolonged time of three hours. The NPG leaf specimens in the present manuscript were prepared by the same process but for a shorter time of half an hour. They probably included silver over 1.2 at. %. The recent investigations have shown that silver on the surface of a NPG leaf plays essential role in the catalytic activity of NPG for the oxidation of CO and the partial oxidation of alcohol (for instance, N. Kamiuchi et al., Nat. Commun. 9 (2018) 2060., L.C. Wang et al., Faraday Discuss. 188 (2016) 57.). However, the present manuscript does not describe even the concentration of residual silver in the NPG leaf specimens.

The authors assumed that methane was decomposed at elevated temperatures by the catalytic surface of NPG leaf specimens. Consequently, an amorphous carbon layer was formed and grown successively on the surface. Inspired by the references in the present manuscript (4, 8, 20-25, 36-38), the authors attempted to draw a conclusion that a bright dot in a HAADF-STEM image (Fig. 1f) in the amorphous carbon layer is attributed to a single gold atom that is moved from the surface of a NPG leaf specimen for the catalytic reaction. First the authors should show that the relevant references and/or the additional experimental data to confirm that NPG is catalytically active for the reaction of CH₄ → C + 2H₂ at elevated temperatures. The references that the authors cited in the manuscript only dealt with a

selective oxidation reaction of methane ($\text{CH}_4 + 1/2 \text{O}_2 \rightarrow \text{CH}_3\text{OH}$) (ref. 10) and a partial oxidation reaction of methane ($\text{CH}_4 + 1/2 \text{O}_2 \rightarrow \text{CO} + 2\text{H}_2$) by not NPG but Ni/CeO₂-ZrO₂ catalysts (ref. 13). Second the authors need to elucidate the following serious questions. 1) Methane may be decomposed into solid carbon by the plasma at the elevated temperature (580 C) and/or electron irradiation in TEM on a metal surface other than the surface of NPG leaf specimens. 2) Even without methane, amorphous carbon (Fig. 1e and Fig. 2) may be induced by electron irradiation on a metal surface as contamination in TEM, 3) Gold atoms (and/or silver atoms) may be removed from the surface of a NPG leaf specimen and may migrate on and/or in the amorphous carbon layer only by the plasma at the elevated temperatures. 4) Islands with lattice fringe in Figs. 2 and 3, especially Fig. 3e may not be pure crystalline gold. The authors should perform chemical analysis by STEM-EDX combined with (S)TEM on these kinds of islands.

The poisoning effect on the surface of catalysts is a central issue in catalysis chemistry and industry. Metals, for instance nickel loses the catalytic activity by the build-up of contamination on the surface. The authors should discuss the poisoning effect on the surface of NPG leaf specimens that are fully covered by thick amorphous carbon in the early stage of the reaction.

Responses to the Comments

Responses to the comments of Reviewer #1:

Comment 1:

In their manuscript "Dynamic co-catalysis of Au single atoms and nanoporous Au for methane pyrolysis" the authors demonstrate the dynamics of methane pyrolysis catalyzed by nanoporous Au observed by in situ TEM. The disaggregation of Au catalyst during the heterogeneous catalytic reaction is observed, which is very important for the understanding of the behavior of nanocatalyst in reactions. They present a careful work with high technical standard and together with the main claim that the released Au single atoms and the nanoporous Au co-catalyze the pyrolysis, it would be appropriate to publish in Nature Communication. However before I can recommend the paper for publication in Nature Communication, I would like the authors to present more results to justify this finding, as there are:

1, The evidence of the existence of Au single atoms.

As described in the manuscript, the Au single atoms can be found in the carbon structures generated by methane pyrolysis. The only evidence of Au single atoms provided is the HAADF image of Fig. 1f. Considering the concept of Au single atom is important to this work. A low magnification HAADF image of an area showing the boundary of nanoporous Au and carbon structure (like Fig. 1e) with a high magnification HAADF image showing the Au single atom of an area from the carbon part are necessarily. In addition and EDX map would be very helpful to understand justify the Au distribution.

Response 1:

Thank you very much for reviewing our manuscript. We greatly appreciate your helpful and constructive comments. According to your suggestion, we have taken low-, medium- and high-magnification HAADF images for an area containing the boundary of nanoporous gold (NPG) and carbon structure, as shown by Fig. R1a–c. Figure R1c and the atomic-resolution HAADF image in Fig. R1d show clearly the Au single atoms in the carbon part of the area. EDS mapping was also performed. As displayed in Fig. R2, the mapping results indicate that the distribution of the Au single atoms in the carbon structure is homogeneous.

The above results have been added into the revised manuscript. They are presented on Pages 5 and 6 of the main text file and Pages 4 and 5 of the Supplementary Information (SI) file, of which Figs. R1 and R2 are added as Supplementary Figs. 3 and 4a,b,d,e. For your convenience, all changes made for the responses have been highlighted by yellow in the main text and the SI files.

Fig. R1 HAADF images of an area with the boundary of NPG and carbon structure. **a–c** Low-, medium- and high-magnification HAADF images of the area. **d** Atomic-resolution HAADF image showing the Au single atoms in the carbon part. Panels **b–d** were taken from the boxed regions in **a–c**, respectively.

Fig. R2 EDS mapping of the Au single atoms in the carbon structure. **a** HAADF image of the Au single atoms. **b,c** EDS mapping images using the Au and the C signals from the region in **a**. **d** EDS spectrum from the region in **a**. The images in **a–c** have the same scale bar.

Comment 2:

2, The disintegration of nanoporous Au.

By comparing the structures of nanoporous Au before and after reaction (Figs 1b and 1d), it is clear that with increasing pore size, the Au areas also increase. That means the nanoporous Au not only disintegrates but also restructures during the reactions. Could you please comment to this fact.

Response 2:

You are absolutely right! Catalysis-induced reconstruction of NPG is an interesting topic that includes many dynamic processes. For instance, *in situ* TEM studies of CO oxidation on NPG (Fujita et al. *Nat. Mater.* 2012, 11, 775; Fujita et al. *Nano Lett.* 2014, 14, 1172) revealed dramatically enhanced surface diffusion of Au which led to substantial coarsening of some pores and some Au areas (namely Au ligaments). In our present case, the coarsening also occurs during the CH₄ pyrolysis, as indicated by Figs. 1b and 1d.

We have used *in situ* TEM to monitor the coarsening, as displayed by Fig. R3. Comparing the circled regions in Figs. R3a and R3b shows that a ligament broke, its residual parts migrated to the adjacent ligament, and they fused together, making the adjacent ligament coarsen. The two papers mentioned above have indicated that the migration and the fusion are caused by the movement of the Au atoms from the broken ligament.

Fig. R3 Catalysis-induced coarsening and breaking of some Au ligaments during the catalytic CH₄ pyrolysis. **a,b** *In situ* TEM images of an NPG sample at 0 s and 75 s after the pyrolysis started at 346 °C. The two red circles have the same position.

The above result and discussion have been added into the revised manuscript. They are presented on Pages 12, 13 of the main text file and Pages 15, 16 of the SI file, of which Fig. R3 is added as Supplementary Fig. 13c,d. The paper of Fujita et al. *Nano Lett.* 2014, 14, 1172 is added into the reference list as ref 43. The paper of Fujita et al. *Nat. Mater.* 2012, 11, 775 was ref 33 in our original manuscript and is ref 34 now due to other revisions.

Comment 3:

3, The role of Au single atoms during reactions.

The co-catalysis of Au single atoms and nanoporous Au is the key idea of this work. The Au single atoms could also catalyze the methane pyrolysis as suggested by the DFT calculation. What is the DFT result if you have little Au clusters maybe composed of 2-3 atoms? However, there is no corresponding experimental evidence for the effect. I understand that experimental evidence is

rather tricky to proof, otherwise you must weaken your statement such as :as 'DFT calculation demonstrates that the released Au single atoms and the nanoporous Au could co-catalyze the pyrolysis...' in the abstract, introduction and conclusion parts.

Response 3:

We are very grateful for your constructive comment. According to your suggestion, we have performed new DFT calculations, as shown by Figs. R4 and R5. Figures R4a and R5 indicate that if an Au cluster composed of 3 atoms (namely Au₃) always exists, the activation energy of its catalyzing the conversion of CH₃* into CH₂* is 3.77 eV. In contrast, if the Au₃ cluster breaks into three Au single atoms under the adsorption of CH₃*, the corresponding activation energy is 0.65 eV (Figs. R4b and R5), much lower than 3.77 eV. This comparison manifests that before the Au₃ cluster can catalyze the conversion of CH₃* into CH₂*, it should have broken into three Au single atoms. The Au₂ cluster has the similar result, as shown by Figs. R4c,d and R5. Then, these resultant Au single atoms catalyze the proceeding and completion of the CH₄ pyrolysis, as indicated by Supplementary Fig. 9 of the SI file.

Fig. R4 DFT-calculated possible evolutions of Au₃ and Au₂ clusters during the CH₄ pyrolysis. **a** The initial and the final states of an Au₃ cluster with the adsorption and decomposition of CH₃* (CH₃ is the first intermediate of the CH₄ pyrolysis) in the case that the Au₃ cluster always exists. CH₃* becomes CH₂* (the second intermediate) in the final state. **b** The initial and the final states of the Au₃ cluster with the adsorption of CH₃* in the case that the final state of the Au₃ cluster is its breaking into three Au single atoms. **c** and **d** have the same meanings as **a** and **b**, but they are for an Au₂ cluster. Grey balls: C in amorphous carbon; cyan balls: C in CH₃* and CH₂*; yellow balls: Au; white balls: H. The C atoms in the amorphous carbon substrate are drawn purposefully to be larger than those in CH₃* and CH₂* in order to distinguish the two types of C atoms.

Fig. R5 Activation energies of the evolutions in Fig. R4. The energy change from an initial state to

its corresponding transition state is the activation energy (E_a).

The above results have been added into the revised manuscript. They are presented on Page 12 of the main text file and Pages 13 and 14 of the SI file, of which Figs. R4 and R5 are added as Supplementary Figs. 11 and 12.

Comment 4:

4, The influence of electron beam.

The dose rate of the e-beam applied for the TEM experiments should be provided. The incident electrons with high energy (200 keV) could also influence the *in situ* experiments. The incident electrons could transfer kinetic energy to the Au nanocatalyst and even sputter the Au atoms on the surface. Thus I would like to request to discuss the possible influence of the electron beam in the manuscript.

Response 4:

We are grateful for your constructive comment. Indeed, the incident high-energy (200 keV) electrons might influence the *in situ* TEM experiments. In our present research, the dose rate of the electron beam applied for the *in situ* experiments is $0.039\text{--}1570\text{ e}\text{\AA}^{-2}\text{s}^{-1}$, which is adjusted by the magnification (4300x–1050000x). To clarify whether the influence of the electron beam is negligible or not, we designed the following experiments.

First, as shown by Fig. R6, under the HRTEM mode with the highest dose rate ($1570\text{ e}\text{\AA}^{-2}\text{s}^{-1}$), 12 min continuous electron beam irradiation at 23 °C did not lead to any observable surface structure change of even a narrow NPG ligament with diameter of $\sim 5\text{ nm}$.

Fig. R6 HRTEM images with the highest electron dose rate before (a) and after 12 min continuous electron beam irradiation (b) without CH_4 . This comparison indicates that the ligament surface structure is stable under the long-time irradiation of electron beam with the highest electron dose rate. The two images have the same scale bar.

Second, *in situ* heating experiment shows that without CH_4 , the structure of nanopores and

ligaments in NPG remained unchanged before the temperature reached 500 °C, as displayed by Fig. R7a–d. Moreover, during this heating process, the HRTEM region was irradiated continuously by the electron beam with the dose rate of $467 \text{ e}\text{\AA}^{-2}\text{s}^{-1}$ for 18 min. From 500 to 600 °C, slight coarsening of nanopores was observed due to surface diffusion enhanced by high temperature (Fig. R7d,e). These results indicate that no observable structure changes occurred in NPG during long-time irradiation of electron beam in heating and below 500 °C.

Fig. R7 Stability of NPG ligaments upon sole electron beam irradiation from 23 to 600 °C in the absence of CH_4 . **a** Low-magnification TEM image at 23 °C. **b–d** HRTEM images of the boxed region in **a** at 23, 346 and 500 °C. **e** Low-magnification TEM image at 600 °C. The images in **a** and **e** have the same scale bar. So do those in **b–d**. The two white boxes in **a** and **e** have the same position.

Third, with atmospheric-pressure pure CH_4 , the structure of NPG underwent substantial change at 346 °C within 75 s (Fig. R8a–d). At high magnification, the boxed region was continuously irradiated by the electron beam (Fig. R8b–d), while other NPG parts were out of the view field and thus not or less affected by the electron beam. When the magnification was lowered (Fig. R8e), the other parts were observed to have structure changes similar to the boxed region. Moreover, comparing these results and those in Figs. R6 and R7 indicates that the structure change of NPG at 346 °C with CH_4 presence is not due to the electron beam, rather, it confirms a catalytic CH_4 pyrolysis process.

Fig. R8 Catalysis-induced structure change of NPG during the CH₄ pyrolysis. **a** Low-magnification TEM image of NPG at 23 °C. The boxed region was thereafter *in situ* characterized from 23 to 346 °C in atmospheric-pressure pure CH₄. **b–d** High-magnification TEM images of the boxed region at 200 °C, 346 °C (0 s) and 346 °C (75 s), respectively. **e** Low-magnification TEM image of NPG after the pyrolysis proceeded at 346 °C for 200 s. The boxed region in **e** has the same position as that in **a**.

Fourth, we used an on-line mass spectrometer to detect hydrogen production when the reaction conditions in TEM were the same as those for Fig. R8 except keeping the electron beam off. The detection results in Fig. R9 indicate that NPG produced hydrogen from CH₄ but a blank chip did not, clearly proving the observed catalytic performance was due to the NPG catalyst itself.

Fig. R9 Hydrogen production results of NPG and a blank chip detected by a mass spectrometer connected to the exhaust end of the *in situ* TEM holder when the electron beam was turned off. These results prove the catalytic role of NPG toward the CH₄ pyrolysis. (The holder uses chips to hold samples, and the blank chip has no NPG samples. Please see details in Methods.)

In summary, all the above results clearly prove that not the electron beam but the NPG-catalyzed

CH₄ pyrolysis plays the dominant role in the NPG structure evolution. The above discussion and details have been added into the revised manuscript. They are presented on Pages 13, 15 and 16 of the main text file and Pages 15–19 of the SI file, of which Figs. R6–R9 are added as Supplementary Figs. 14, 15, 13 and 16, respectively.

Comment 5:

5, The pore size distribution.

The samples are treated by a mixture of 15 ml HCl (37%) and 5 ml HNO₃ (65%) before the measurement of pore size distribution. Why you removed the nanoporous Au? And will the strong oxidizing mixture solution influence the structure of amorphous carbon? In addition, nitrogen adsorption-desorption measurement is only suitable for studying the nanopores of carbon structure. It might be advised to use CO₂ adsorption-desorption measurement to study the sub-nano pores of carbon structure.

Response 5:

Thank you for this valuable comment. According to your suggestion, we carried out CO₂ and N₂ adsorption-desorption measurements to study the sub-nano pores and nanopores of the carbon layers while keeping the NPG substrates. The results are shown by Figs. R10 and R11. Just as you predicted, Fig. R10 clearly reveals a sub-nano pore feature of the carbon layers. The sub-nano pores have sizes centered at 0.82 nm. Figure R11 shows that the nanopores in the carbon layers have sizes of 3–7 nm.

Fig. R10 Size distribution of sub-nano pores in the amorphous carbon layers, detected by CO₂ adsorption-desorption.

Fig. R11 Size distribution of nanopores in the amorphous carbon layers, detected by N_2 adsorption-desorption.

The above discussion and details have been added into the revised manuscript. They are presented on Pages 8, 9 and 15 of the main text file. The N_2 adsorption-desorption result of Fig. 3a has been replaced by Fig. R11, and Fig. R10 is added as Supplementary Fig. 6 in Page 8 of the SI file.

Comment 6:

6. Technical details

Fig. 1c is highly defocused. Please provide an in-focus image. In general, the Fourier transform pattern of a HRTEM image shows more clearly all the frequencies present in the image, then just to mark selected lattice fringe spacings in the image.

Response 6:

We appreciate very much your careful checking. According to your suggestion, we have taken an in-focus HRTEM image and its fast Fourier transform pattern, as shown by Fig. R12. Figure 1c has been replaced by Fig. R12, and please see it on Page 5 of the main text file.

Fig. R12 In-focus HRTEM image and its fast Fourier transform pattern of NPG.

Responses to the comments of Reviewer #3:

Comment 1:

The authors attempted to visualize the role of individual Au atoms played in the reaction with methane on the surface of a nanoporous Au (NPG) leaf specimen by applying in-situ transmission electron microscopy. The challenge to push in-situ observation forward is highly important and useful for elucidating the catalytic reactions in real space and time. Therefore, any study as this manuscript potentially deserves for publication. After reading this manuscript, it is however concluded that the experiments by the authors were not well organized and the analyses were vague even though the title of the manuscript appeared to be fascinating. The readership of Nature Communications will be misled by the manuscript.

In any NPG leaf specimen that is prepared by the dealloying method, residual silver inevitably remains over 1 at.%. Fujita et al. already reported in their paper (Nat. Mater. 11 (2012) 775.) that silver was included in a NPG leaf specimen by 1.2 at. % after the dealloying process even for a prolonged time of three hours. The NPG leaf specimens in the present manuscript were prepared by the same process but for a shorter time of half an hour. They probably included silver over 1.2 at.%. The recent investigations have shown that silver on the surface of a NPG leaf plays essential role in the catalytic activity of NPG for the oxidation of CO and the partial oxidation of alcohol (for instance, N. Kamiuchi et al., Nat. Commun. 9 (2018) 2060., L.C. Wang et al., Faraday Discuss. 188 (2016) 57.). However, the present manuscript does not describe even the concentration of residual silver in the NPG leaf specimens.

Response 1:

We appreciate very much your interest in our research and your valuable comments. The discussion on the silver content and its possible role will indeed enrich this work. We first measured the residual Ag content in our NPG sample, which is 1.37 ± 0.38 at% (Fig. R13), and the Ag distribution in NPG is homogeneous (Fig. R14).

Fig. R13 SEM images (a,b) and EDS result (c) of the as-prepared NPG sample used for the original version of our manuscript. From 8 EDS measurements, the average Ag content is found to be 1.37 at%, and the corresponding standard deviation is 0.38 at%.

Fig. R14 EDS mapping of as-prepared NPG. **a** HAADF image. **b,c** Maps of Au and Ag. **d** Overlay of **a–c**. The images in **a–d** have the same scale bar.

In order to evaluate the role of the residual Ag in the CH₄ pyrolysis, we fabricated a series of new NPG samples with different Ag contents and connected a mass spectrometer to the exhaust end of the *in situ* TEM holder for detecting hydrogen production while keeping the electron beam off. The results are shown in Fig. R15. Obviously, varying the Ag content causes different activities, and in particular the NPG sample with the least Ag content (*the catalyst discussed in our original manuscript*) shows a distinct behavior where it becomes active only when the reaction temperature reaches 200 °C and higher. In contrast, the NPG samples with the higher Ag contents start to exhibit activities at around 50 °C.

Fig. R15 Hydrogen production results during the CH₄ pyrolysis reaction using the NPG samples with different Ag contents and a blank chip without NPG. The electron beam was turned off during these experiments. The smooth lines are for eye guide. (The *in situ* TEM holder uses chips to hold samples, and the blank chip has no NPG samples. Please see details in Methods.)

As the reviewer kindly suggested that the residual Ag might contribute to the catalytic activity of NPG samples, our above observations indeed prove his/her prediction. Comparisons from different samples show that with the decrease of the Ag content, the low-temperature (<200 °C) activity decreases remarkably, and for our discussed NPG sample with the lowest Ag content, it shows CH₄ pyrolysis activity only above 200 °C. Actually, we selected this particular sample (with the lowest Ag content) because previous reports have proven that CH₄ can be pyrolyzed by pure gold (Lang et al. *Angew. Chem. Int. Ed.* 2017, 56, 13406). In our present work, we are not searching for better-activity catalysts, rather, we use the NPG (with the lowest Ag content) as a nice model system to disclose a highly dynamic process of CH₄ pyrolysis, catalyst restructuring (especially Au single atom formation), and a unique co-catalysis process. The drastic structural change of NPG catalysts provides clear evidence of how Au ligaments restructure (Fig. 2 and Supplementary Movies 1 and 2) and how Au single atoms form, aggregate and re-disintegrate (Fig. 3 and Supplementary Movies 4 and 5). It is quite unlikely for such low Ag content to cause such drastic change processes, and no Ag signals were detected in the carbon layers (please see details in Response 6). Therefore, for all the drastic change processes, Au reasonably plays the dominant role, and these observations in an important C1 conversion system, as you suggested in your comments, "is highly important and useful for elucidating the catalytic reactions in real space and time. Therefore, any study as this manuscript potentially deserves for publication".

We value very much your constructive comments, and the above discussion and details have been added into the revised manuscript. They are presented on Pages 5, 6, 13, 14 and 15 of the main text file and Pages 2, 3, 20 and 21 of the Supplementary Information (SI) file, of which Figs. R13–R15 are added as Supplementary Figs. 1, 2 and 17, respectively. The paper of Lang et al. *Angew. Chem. Int. Ed.* 2017, 56, 13406 is cited as ref 14. For your convenience, all changes made for the responses have been highlighted by yellow in the main text and the SI files.

Comment 2:

The authors assumed that methane was decomposed at elevated temperatures by the catalytic surface of NPG leaf specimens. Consequently, an amorphous carbon layer was formed and grown successively on the surface. Inspired by the references in the present manuscript (4, 8, 20-25, 36-38), the authors attempted to draw a conclusion that a bright dot in a HAADF-STEM image (Fig. 1f) in the amorphous carbon layer is attributed to a single gold atom that is moved from the surface of a NPG leaf specimen for the catalytic reaction. First the authors should show that the relevant references and/or the additional experimental data to confirm that NPG is catalytically active for the reaction of CH₄ → C + 2H₂ at elevated temperatures. The references that the authors cited in the manuscript only dealt with a selective oxidation reaction of methane (CH₄ + 1/2 O₂ → CH₃OH) (ref. 10) and a partial oxidation reaction of methane (CH₄ + 1/2O₂ → CO + 2H₂) by not NPG but Ni/CeO₂-ZrO₂ catalysts (ref. 13).

Response 2:

Thank you for this great suggestion. The Landman group has published many seminal researches on C–H bond cleavage by Au catalysts, and in particular, their paper (Lang et al. *Angew. Chem. Int. Ed.* 2017, 56, 13406) provided nice experimental and simulation results of how Au pyrolyzed CH₄.

On the other hand, our mass spectrometer detected hydrogen production which confirmed the catalytic role of NPG for the reaction of $\text{CH}_4 \rightarrow \text{C} + 2\text{H}_2$ at elevated temperatures, as shown by Fig. R16, even when the electron beam was turned off.

Fig. R16 Hydrogen production results of the discussed NPG sample and the blank chip detected by mass spectrometer, proving the catalytic role of NPG toward the CH_4 pyrolysis reaction even when the electron beam was turned off. The two curves are from Fig. R15.

The above discussion and details have been added into the revised manuscript. They are presented on Page 13 of the main text file and Page 19 of the SI file, of which Fig. R16 is added as Supplementary Fig. 16. The paper of Lang et al. *Angew. Chem. Int. Ed.* 2017, 56, 13406 is cited as ref 14.

Comment 3:

Second the authors need to elucidate the following serious questions. 1) Methane may be decomposed into solid carbon by the plasma at the elevated temperature (580 C) and/or electron irradiation in TEM on a metal surface other than the surface of NPG leaf specimens.

Response 3:

We understand that the possible role of electron beam irradiation may interfere with the results in many *in situ* TEM studies.

In our present *in situ* study, we did not use any plasma to treat our samples, and the typical reaction temperature was kept at 346 °C or below. All NPG samples were prepared by the standard dealloying method (free corrosion at 30 °C), followed by rinsing and drying. No other special *ex situ* treatments were used on them before they were inserted into TEM for the *in situ* study. As to your question regarding the possible role of electron beam, its effect has been ruled out in our second response to your comments. That is, the on-line mass spectrometer detected hydrogen production, confirming the catalytic role of NPG for the reaction of $\text{CH}_4 \rightarrow \text{C} + 2\text{H}_2$ at elevated temperatures, as shown by Fig. R16, even when the electron beam was turned off. In addition, the plasma experiment outside TEM is used to rapidly obtain a mass of carbon-loaded NPG samples, whose structures are shown in Fig. 1d of the main text file and quite similar to those of the samples through the *in situ* study (Fig. R17b).

Fig. R17 Catalysis-induced structure change of NPG during the CH₄ pyrolysis. **a** TEM image of an NPG sample at 23 °C. **b** TEM image of the sample after the pyrolysis proceeded at 346 °C for 7 min. The two images have the same scale bar.

The above discussion and details have been added into the revised manuscript. They are presented on Pages 13 and 15 of the main text file and Page 22 of the SI file, of which Fig. R17 is added as Supplementary Fig. 18.

Comment 4:

2) Even without methane, amorphous carbon (Fig. 1e and Fig. 2) may be induced by electron irradiation on a metal surface as contamination in TEM,

Response 4:

It is true that electron irradiation may sometimes induce amorphous carbon contamination. In our work, the as-prepared NPG samples and the TEM chamber are both clean. As shown by Fig. R18, the surface of NPG ligaments remains clean upon continuous 12 min electron irradiation under the HRTEM mode, without feeding the CH₄ gas. Besides, in our second response with Fig. R16, we clearly prove that NPG can pyrolyze CH₄ to produce hydrogen without electron beam, indicating that the amorphous carbon on the NPG surface is induced by the CH₄ pyrolysis.

Fig. R18 HRTEM images before (a) and after 12 min continuous electron beam irradiation (b) without the CH₄ pyrolysis. The two images have the same scale bar.

The above discussion and details have been added into the revised manuscript. They are presented on Page 13 of the main text file and Page 17 of the SI file, of which Fig. R18 is added as

Supplementary Fig. 14.

Comment 5:

3) Gold atoms (and/or silver atoms) may be removed from the surface of a NPG leaf specimen and may migrate on and/or in the amorphous carbon layer only by the plasma at the elevated temperatures.

Response 5:

As mentioned above, our samples for the *in situ* study were prepared via the standard dealloying method and were not treated with any special method such as plasma treatment. After the *in situ* TEM experiments, the samples were characterized by HAADF, showing clearly the existence of Au single atoms in the carbon layers (Fig. R19). This result indicates that the Au single atoms were produced by the disintegration of the NPG surface during the CH₄ pyrolysis. Moreover, we have also prepared similar NPG samples with carbon via the *ex situ* CH₄ pyrolysis reaction. These samples were also confirmed to contain Au single atoms (Fig. 1f). By comparing both *in situ* (with electron irradiation) and *ex situ* (without electron irradiation) results, we reveal a highly dynamic process of how Au catalyzes the CH₄ pyrolysis in real space and time.

Fig. R19 Characterization of Au single atoms produced by the *in situ* CH₄ pyrolysis. **a** Low-magnification HAADF image of an NPG sample after it experienced the *in situ* CH₄ pyrolysis. **b** Atomic-resolution HAADF image of a carbon region of the sample, showing clearly Au single atoms.

The above discussion and details have been added into the revised manuscript. They are presented on Page 15 of the main text file and Page 23 of the SI file, of which Fig. R19 is added as Supplementary Fig. 19.

Comment 6:

4) Islands with lattice fringe in Figs. 2 and 3, especially Fig. 3e may not be pure crystalline gold. The authors should perform chemical analysis by STEM-EDX combined with (S)TEM on these kinds of islands.

Response 6:

STEM-EDS (namely STEM-EDX) and lattice fringe measurements have been carried out, as shown below by Fig. R20, which provide indisputable evidences that these nanoparticles are pure crystalline Au.

Fig. R20 HAADF images and their EDS results of some Au nanoparticles from an NPG sample through the CH₄ pyrolysis. **a–c** Low-magnification and atomic-resolution HAADF images of the Au nanoparticles. The images in **b** and **c** were taken from the boxed regions in **a**. Both of the lattice fringe spacings and the angles between the fringes conform to the standard Au structure, indicating that the nanoparticles are crystalline Au. **d–i** EDS results of these Au nanoparticles. The Ag signals in **i** are below the detecting limit, and only noise exists in the Ag map in **f**, reasonably suggesting that there is no Ag in these Au nanoparticles and the carbon layers. The images in **a** and **d–g** have the same scale bar.

Furthermore, we performed the EDS mapping of the amorphous carbon layers containing Au single atoms. The result shows that no Ag signals were detected, either (Fig. R21). That is, the single atoms in the amorphous carbon layers are Au. Thus, the nanoparticles formed by the aggregation of the Au single atoms are confirmed to be pure Au again.

Fig. R21 EDS mapping of the Au single atoms in the carbon structure. **a** HAADF image of the Au single atoms. **b–d** EDS mapping images using the Au, Ag and C signals from the region in **a**. **e,f** EDS spectra from the region in **a**. The images in **a–d** have the same scale bar. The mapping results indicate that the distribution of the Au single atoms in the carbon structure is homogeneous. The Ag signals in **f** are below the detecting limit, and only noise exists in the Ag map in **c**, reasonably suggesting that there is no Ag in the carbon layer.

The above discussion and details have been added into the revised manuscript. They are presented on Page 10 of the main text file and Pages 5 and 9 of the SI file, of which Figs. R20 and R21 are added as Supplementary Figs. 7 and 4.

Comment 7:

The poisoning effect on the surface of catalysts is a central issue in catalysis chemistry and industry. Metals, for instance nickel loses the catalytic activity by the build-up of contamination on the surface. The authors should discuss the poisoning effect on the surface of NPG leaf specimens that are fully covered by thick amorphous carbon in the early stage of the reaction.

Response 7:

Thank you for your constructive comment. Within the scope of our present study, NPG catalyzes CH₄ decomposition in a highly dynamic way. The deposition of amorphous carbon layers is accompanied by tremendous H₂ gas evolution, which results in the formation of multi-modal porous structure in the carbon layers. In order to characterize this unique structural feature, we carried out CO₂ and N₂ gas adsorption-desorption measurements (Figs. R22 and R23) which demonstrated the existence of both sub-nano pores and nanopores in the carbon layers. These pores can serve as efficient channels for mass transportation, which are key to the observed co-catalysis behavior in our system. Actually, Fig. 2 of the main text file indicates that after the NPG surface was fully covered by amorphous carbon layers, the surface still continuously disintegrated, making NPG ligaments continuously slim. This phenomenon manifests that the pores truly worked as efficient channels for mass transportation and the catalytic reaction still occurred on the NPG

surface that was covered by the amorphous carbon. That is, the pores effectively alleviated the poisoning effect that normally occurred on the catalyst surface.

Fig. R22 Size distribution of sub-nano pores in the amorphous carbon layers, detected by CO_2 adsorption-desorption.

Fig. R23 Size distribution of nanopores in the amorphous carbon layers, detected by N_2 adsorption-desorption.

The above discussion and details have been added into the revised manuscript. They are presented on Pages 8 and 9 of the main text file and Page 8 of the SI file. The original version of Fig. 3a has been replaced by Fig. R23, and Fig. R22 is added as Supplementary Fig. 6.

Reviewers' comments:

Reviewer #1 (Remarks to the Author):

Comment to Response 1,

The new results about the HAADF images and EDS mapping are sufficient to characterize the Au single atoms.

Comment to Response 2,

The additional results nicely answer my concern about the reconstruction of NPG.

Comment to Response 3,

I am satisfied with the DFT calculation about the Au₃ and Au₂. But I still suggest weakening your statement of the 'co-catalysis of Au single atoms and nanoporous Au' in the abstract and conclusion, such as :as demonstrated by DFT calculation, the Au single atoms could co-catalyze the reaction with nanoporous Au.'

Comment to Response 4,

The new experiments are sufficient to prove electron beam is not the main inducement for the reconstruction of NPG during catalytic reaction.

Comment to Response 5,

The new CO₂ and N₂ adsorption-desorption measurements are carried out nicely as suggested.

Comment to Response 6,

The new HRTEM image is acceptable.

Reviewer #3 (Remarks to the Author):

Thank you for consideration of my previous concerns.

First I appreciate the authors agreeing that your sample are not a NPG leaf of pure crystalline gold but a NPG leaf of Au-Ag alloy as in the previous studies (Response 1). Any impurities atoms (such as Ag in this

study) become more moveable at higher temperature. In Fig. R15, the authors described that “the NPG sample with the least Ag content (the catalyst discussed in our original manuscript) shows a distinct behavior where it becomes active only when the reaction temperature reaches 200C and higher. In contrast, the NPG sample with the higher Ag content start to exhibit activities at around 50C.” This result is simply accounted for by assuming that surface Ag atoms contribute to catalysis: At higher temperature (over 200 C), Ag atoms become more moveable. Ag atoms of even smaller amount (1.7at % Ag) can reach the surface of a NPG leaf for the catalytic reaction. Ag atoms of larger amount (over 9.3at.%Ag) can be exposed on the surface even at lower temperature at 50C. The authors also intend to claim that residual Ag does not exist on the surface of NPG by electron microscopy analysis (STEM-EDS) on the nanoparticles from an NPG sample (Response 6). However, residual Ag atoms of small concentration embedded in the nanoparticles could not be identified by these kind of analysis. By in-situ electron microscopy, Kamiuchi et al. (Nature Comm. 9, 2060 (2018)) carefully measured the distance between adjacent atomic columns (not lattice planes) on the surface of a NPG leaf during a catalytically chemical reaction and could show that there are residual Ag atoms for the catalysis. Hence the authors fail to confirm that Ag atoms in the NPG leaf never contribute to the catalytic activity. The authors should revise the manuscript by adding the discussion on the point above and referring the previous study fairly for the readership of Nature Communication.

Second I appreciate again the authors confirming that the catalytically chemical reaction is occurring without electron irradiation (Response 2). However, the main claim in the previous and revised manuscript is in-situ transmission electron microscopy on individual surface Au atoms. The authors concluded that the Au atoms are moveable (departing from and agglomerating into crystalline gold) upon catalysis without electron irradiation. However, there is no experimental evidence to rule out the migration of Au atoms that is induced by electron irradiation. The electron microscopy evidence that the authors provided us with in the revised manuscript shows only the very blurred (defocused) futures on the surface of the specimen not at atomic scale but with 2nm in accuracy (Fig. R18). Hence, my concern remains. The in-situ electron microscopy data is induced essentially by electron irradiation regardless of catalyst.

As a result, I am not able to recommend the revised manuscript in publication in Nature Communication unless the authors can address the comments and the criticisms above.

Responses to the Comments

Responses to the comments of Reviewer #1:

Comments:

Comment to Response 1,

The new results about the HAADF images and EDS mapping are sufficient to characterize the Au single atoms.

Comment to Response 2,

The additional results nicely answer my concern about the reconstruction of NPG.

Comment to Response 3,

I am satisfied with the DFT calculation about the Au₃ and Au₂. But I still suggest weakening your statement of the 'co-catalysis of Au single atoms and nanoporous Au' in the abstract and conclusion, such as: as demonstrated by DFT calculation, the Au single atoms could co-catalyze the reaction with nanoporous Au.'

Comment to Response 4,

The new experiments are sufficient to prove electron beam is not the main inducement for the reconstruction of NPG during catalytic reaction.

Comment to Response 5,

The new CO₂ and N₂ adsorption-desorption measurements are carried out nicely as suggested.

Comment to Response 6,

The new HRTEM image is acceptable.

Responses:

We greatly appreciate your approval and your constructive comments on our work. According to your advice, the statements of "co-catalysis of Au single atoms and nanoporous Au" in the abstract and conclusion have been replaced by "as demonstrated by DFT calculation, the Au single atoms could co-catalyze the reaction with nanoporous Au". They are presented on Pages 2 and 14 of the main text file. For your convenience, all changes made for the responses have been highlighted by yellow in the main text file.

Responses to the comments of Reviewer #3:

Comment 1:

Thank you for consideration of my previous concerns.

First I appreciate the authors agreeing that your sample are not a NPG leaf of pure crystalline gold but a NPG leaf of Au-Ag alloy as in the previous studies (Response 1). Any impurities atoms (such as Ag in this study) become more moveable at higher temperature. In Fig. R15, the authors described that "the NPG sample with the least Ag content (the catalyst discussed in our original manuscript) shows a distinct behavior where it becomes active only when the reaction temperature reaches 200C and higher. In contrast, the NPG sample with the higher Ag content start to exhibit activities at around 50C." This result is simply accounted for by assuming that surface Ag atoms contribute to catalysis: At higher temperature (over 200 C), Ag atoms become more moveable. Ag atoms of even smaller amount (1.7at % Ag) can reach the surface of a NPG leaf for the catalytic reaction. Ag atoms of larger amount (over 9.3at.%Ag) can be exposed on the surface even at lower temperature at 50C. The authors also intend to claim that residual Ag does not exist on the surface of NPG by electron microscopy analysis (STEM-EDS) on the nanoparticles from an NPG sample (Response 6). However, residual Ag atoms of small concentration embedded in the nanoparticles could not be identified by these kind of analysis. By in-situ electron microscopy, Kamiuchi et al. (Nature Comm. 9, 2060 (2018)) carefully measured the distance between adjacent atomic columns (not lattice planes) on the surface of a NPG leaf during a catalytically chemical reaction and could show that there are residual Ag atoms for the catalysis. Hence the authors fail to confirm that Ag atoms in the NPG leaf never contribute to the catalytic activity. The authors should revise the manuscript by adding the discussion on the point above and referring the previous study fairly for the readership of Nature Communication.

Response 1:

We are very grateful for your helpful and constructive comments. By reading them, we think that some descriptions in our previous submissions may be not clear, which might have caused some misunderstanding. Actually, we have never thought or claimed that "Ag atoms in the NPG leaf never contribute to the catalytic activity". On the contrary, we agree with your viewpoint that Ag atoms in the NPG leaf contribute to the catalytic activity. As you pointed out, our control experiments using NPG samples with varied residual Ag contents clearly showed that Ag atoms are capable of improving the catalyst performance and thus contribute to the catalytic activity.

In our present study, the key finding is a highly dynamic structure evolution process of a nanostructured catalyst during operation. We demonstrated that NPG ligaments containing massive atoms may rapidly disintegrate during the catalytic methane pyrolysis process at 346 °C. The ligament dimension could remarkably decrease from around 13 nm to 5 nm within 2 seconds (Fig. 2d–f in the main text). Considering that the residual Ag content is relatively low (1.37 ± 0.38 at%, whose detailed data are given in the Source Data file) in our sample, it is therefore reasonable to conclude that Au plays a dominant role in this drastic structural evolution process involving massive atoms. This structural evolution includes ligament disintegration, Au single atoms formation, and Au nanoparticle formation.

We agree that heterogeneous doping/alloying of other elements such as Ag into Au may modulate Au's catalytic properties, and in some reactions their roles may become dominant, such as CO oxidation as emphasized by Kamiuchi et al. (Nature Comm. 9, 2060 (2018)). In our present case,

however, this is not the key issue for the following two reasons. First, it has been well acknowledged that pure Au is capable for catalytic C-H activation and CH₄ pyrolysis, based on both experimental and theoretical studies (e.g. refs 9 and 14), although its real-time structure evolution process during service has never been observed (which is the contribution of our present work). Second, based on your suggestions, we have supplemented the same methane pyrolysis experiments using pure Au nanoparticles (Fig. R1), and here these pure Au nanoparticles were produced by magnetron sputtering with a pure Au (>99.99%) target. Our in-situ TEM and hydrogen production studies clearly proved that these pure Au nanoparticles can catalyze the CH₄ pyrolysis under the same reaction conditions. Therefore, in our work, it is reasonable to use the NPG sample with the lowest Ag content to monitor its structural evolution because the effect of Ag may be minimized.

Fig. R1 Characterization of methane pyrolysis catalyzed by pure Au nanoparticles that were produced by magnetron sputtering. **a–c** In-situ observation of the methane pyrolysis at 346 °C. The three images in **a–c** have the same scale bar. In the first 90 seconds, there was no methane in the system, and the Au nanoparticles did not change in morphology. After the methane was introduced for 12 seconds, the morphology of the Au nanoparticles changed and small nanoparticles appeared, as indicated by the white arrows in **c**. **d** Mass spectrometry (MS) result of the hydrogen production from the methane pyrolysis catalyzed by the Au nanoparticles at 346 °C when the electron beam was turned off. The moment of introducing methane is at 330 seconds, as indicated by the black arrow. This MS result indicates that H₂ was produced by CH₄ and the Au nanoparticles. In contrast, Supplementary Fig. 16 in the Supplementary Information (SI) file has shown that when CH₄ was introduced into a blank chip without Au nanoparticles, no H₂ was produced under the same experimental conditions. Thus, the production experiments of hydrogen and the morphology change of the pure Au nanoparticles prove that the pure Au nanoparticles have the ability to catalyze the methane pyrolysis.

The STEM-EDS results in Response 6 in the first round of our revision (now displayed as Supplementary Fig. 7 of the SI file) show that Au nanoparticles evolved from the NPG surface disintegration do not contain Ag. But we do not intend to use this finding to claim that residual Ag does not exist on the surface of NPG. On the contrary, we agree that Ag exists on the NPG surface, because of the following facts: these NPG-evolved Au nanoparticles are separate from and do not contact NPG surfaces, as shown by Fig. 3e in the main text and Supplementary Figs. 7 and 8 (for instance, the substrate containing the Au nanoparticles in Supplementary Fig. 7 is not NPG but amorphous carbon), although the nanoparticles were formed by Au single atoms that were produced from the NPG surface disintegration; Ag atoms in the NPG leaf contribute to the catalytic activity, as mentioned in the first paragraph of this response. We appreciate very much your kind sharing of Kamiuchi et al.'s work (Nature Comm. 9, 2060 (2018)), which has been added into the reference list as ref 37 (Page 22 of the main text file). This work confirmed the existence of surface Ag by measuring the atomic spacing of the NPG surfaces, in which the NPG surfaces have no amorphous carbon on them. But our NPG-evolved Au nanoparticles are in amorphous carbon,

causing that our work cannot use the method of Kamiuchi et al. to detect Ag atoms. The STEM-EDS method we used is capable of detecting Ag with the content of 0.87 at% or higher in NPG (please see the Source Data file), and by it no Ag signals were detected in our NPG-evolved Au nanoparticles. This result indicates that the Ag content in our Au nanoparticles is below the detecting limit of EDS. Your comments make us realize that we should state this finding more rigorously, and so we have revised the statement on our NPG-evolved Au nanoparticles to "the Ag content in these crystalline Au nanoparticles is below the detecting limit of EDS".

Our descriptions and statements related to the above results and discussions have been revised to clarify our findings and emphasize the contribution of residual elements such as Ag. The related revisions including the citations of ref 37 are presented on Pages 4, 6, 13, 14 and 15 of the main text file and Pages 9, 10, 25, 26 and 27 of the SI file, of which Fig. R1 is added as Supplementary Fig. 21. For your convenience, all changes made for the responses have been highlighted by yellow in the main text and the SI files.

Comment 2:

Second I appreciate again the authors confirming that the catalytically chemical reaction is occurring without electron irradiation (Response 2). However, the main claim in the previous and revised manuscript is in-situ transmission electron microscopy on individual surface Au atoms. The authors concluded that the Au atoms are moveable (departing from and agglomerating into crystalline gold) upon catalysis without electron irradiation. However, there is no experimental evidence to rule out the migration of Au atoms that is induced by electron irradiation. The electron microscopy evidence that the authors provided us with in the revised manuscript shows only the very blurred (defocused) futures on the surface of the specimen not at atomic scale but with 2nm in accuracy (Fig. R18). Hence, my concern remains. The in-situ electron microscopy data is induced essentially by electron irradiation regardless of catalyst.

As a result, I am not able to recommend the revised manuscript in publication in Nature Communication unless the authors can address the comments and the criticisms above.

Response 2:

We appreciate very much your valuable comments. For your concern about the quality of our TEM images, we have another pair of TEM images with higher resolution, as displayed in Fig. R2. Comparing the two images indicates that the NPG ligament surface structure is clean and stable under the 5 min irradiation of electron beam with the highest electron dose rate.

In addition, we have performed new high-resolution TEM (HRTEM) experiments to provide a comparison of the NPG structures with and without the methane pyrolysis catalysis at 346 °C when the electron beam was kept on (Figs. R3 and R4). Figure R3 clearly shows that once CH₄ was introduced at 346 °C, the NPG surface structure changed violently within 0.25 seconds, implying that the violent migration of Au atoms from the NPG surface took place within 0.25 seconds. In contrast, Fig. R4 clearly shows that when the electron beam irradiation was kept on with no CH₄ introduction, the NPG ligament surface structure was clean and stable at 346 °C for 68 seconds, which is much longer than 0.25 seconds.

Besides, Egerton and co-workers have provided serious discussions regarding the role of high-energy electron beam (Egerton et al. Ultramicroscopy 110, 991 (2010)), indicating that the minimum incident-electron energy to knock out Au atoms is approximately 407 keV, much higher than 200 keV

of the electron irradiation used in our work.

Therefore, we can conclude that the migration of Au atoms from the NPG surfaces in the in-situ methane pyrolysis of our work is not induced by the electron irradiation, rather it is due to the CH₄ pyrolysis catalysis.

Fig. R2 HRTEM images before (a) and after 5 min continuous electron beam irradiation (b) without the CH₄ pyrolysis at 23 °C. The two images have the same scale bar. The electron dose rate used to take the images is the highest used in our experiments (1570 eÅ⁻²s⁻¹).

Fig. R3 HRTEM images at 0 second (a) and 0.25 seconds (b) during the CH₄ pyrolysis at 346 °C. The

two images have the same scale bar. The electron beam was kept on. The electron dose rate used to take the images is the highest used in our experiments ($1570 \text{ e}\text{\AA}^{-2}\text{s}^{-1}$).

Fig. R4 HRTEM images before (a) and after 68-second continuous electron beam irradiation (b) without the CH_4 pyrolysis at $346 \text{ }^\circ\text{C}$. The two images have the same scale bar. The electron dose rate used to take the images is the highest used in our experiments ($1570 \text{ e}\text{\AA}^{-2}\text{s}^{-1}$). The irradiation duration of 68 seconds is much longer than 0.25 seconds of Fig. R3.

The above results and discussions have been added into the revised manuscript. The paper of Egerton et al. *Ultramicroscopy* 110, 991 (2010) is added into the reference list as ref 45 (Page 23 of the main text file). The related revisions are presented on Page 13 of the main text file and Pages 21, 22 and 23 of the SI file, of which Figs. R2, R3 and R4 are added as Supplementary Figs. 17, 18 and 19, respectively.

Reviewers' comments:

Reviewer #4 (Remarks to the Author):

On studying the manuscript and supplemental data, I have some additional concerns to those raised by reviewers 1 and 3, which the authors should consider.

What are the white patches seen in the HAADF image Fig 1f? Could they be 2D Au thin rafts? They have similar dimensions to the 'particles' noted later in the manuscript.

There could be Ag atoms also present in Fig 1f - they will show significant mass contrast (albeit less than the Au) against the C support.

The intensity of the individual atom images and their clarity/width will depend on their height in the sample (i.e., whether they are on the top surface, in the interior, or on the bottom surface of the C). Were systematic through focal HAADF series taken to investigate the depth distribution of the atomic species? How does the visibility of the white patchy areas change with defocus?

I am not entirely convinced the 'particles' shown in movies 4 and 5 are actually particles (- they look more like thin Au rafts).

I am also unconvinced that the particles in movies 4 and 5 are 'disintegrating'. They could just be re-orienting as the C film grows and moves. In fact, in Movie 5 it looks like the 200-type 'raft' like fringes disappear then later re-appear in approximately the same location as 111-type fringes.

Much of what is occurring at the Au/C interface and on the Au 'clean' surfaces is obscured by the strong delocalization effects (i.e., fringes extending beyond the edge of the gold) due to the FEG electron source. The authors should comment on this.

Even with 1.4 ± 0.4 wt% Ag present (as measured by EDS), at least 1 in 100 of the dispersed atoms will still be Ag. It is also qualitatively noted that the catalytic activity decreases with decreasing Ag content. Does it drop linearly? The fact that the activity measurably changes with Ag content in this way suggests that they are indeed quite active species. If the Ag atoms are much more active compared to the Au, then it cannot be simply claimed that the behavior is dominated by the Au, and that the Ag contribution is insignificant.

Some of the terminology used by the authors (e.g violently changing, massive) while being very evocative is not very scientific.

Responses to the Comments

Responses to the comments of Reviewer #4:

On studying the manuscript and supplemental data, I have some additional concerns to those raised by reviewers 1 and 3, which the authors should consider.

Comment 1:

What are the white patches seen in the HAADF image Fig 1f? Could they be 2D Au thin rafts? They have similar dimensions to the 'particles' noted later in the manuscript.

Response 1:

We are very grateful for your helpful comments. Your suggestion about the structure of white patches in Fig. 1f is very reasonable, and it gives us great inspiration. This work reveals that during the methane pyrolysis, the NPG catalyst undergoes very dramatic structural evolution, during which various Au nano- or subnano-structures are dynamically observed, including Au single atoms, Au atom pairs, Au clusters (Supplementary Figs. 3c, d, 5a, b, 14), Au nanoparticles (Fig. 3e and Supplementary Figs. 11–14), and even 2D Au thin rafts, etc.

According to your suggestions, we re-examined more than two hundred HAADF images recorded during our research, and found that 2D Au rafts indeed existed, although they were quite rare due to their metastability. Figure R1 shown below features the co-existence of Au single atoms, Au clusters, and a possible 2D Au raft. From the three-dimensional (3D) contrast intensity distribution of the Au structures (Fig. R1b), it can be clearly seen that the contrast intensity of the main part of the Au raft is approximately uniform, while the contrast intensities in the central parts of the Au clusters are much higher than those of their edges.

Fig. R1 **a** HAADF image featuring the co-existence of Au single atoms, clusters and 2D thin raft. **b** 3D rendering of the contrast intensity distribution of **a**.

The above results and analysis have been added into the revised manuscript. They are presented on Page 11 of the main text file and Page 19 of the Supplementary Information (SI) file, of which Fig. R1 is added as Supplementary Fig. 14. For your convenience, all changes made for the responses have been highlighted by yellow in the main text, the SI and the Supplementary Movie note files.

Comment 2:

There could be Ag atoms also present in Fig 1f - they will show significant mass contrast (albeit less than the Au) against the C support.

Response 2:

Thank you for your helpful comment. The content of Ag in the original NPG catalyst is 1.37 ± 0.38 at%, so there might be Ag atoms somewhere in the amorphous carbon in Fig. 1f. We performed EDS on the amorphous carbon supports (Supplementary Figs. 4 and 11), and found Au signals were obvious, whereas, no Ag signals were detected, indicating that the Ag contents, if exist in the supports, were very low. Indeed, Fig. 1f shows that some single atoms have higher HAADF contrast intensities than the others. However, the contrast intensity of a single atom is related to its distance from the focal plane (see **Response 3** for detailed analysis). We cannot simply assume that certain bright spots with lower intensities in Fig. 1f are Ag single atoms. We supplemented some new experiments to analyze the contrast difference between Au and Ag single atoms in a same focal plane. The following experimental results show obvious contrast difference between Au and Ag single atoms.

Firstly, we used the magnetron sputtering technique to prepare a new sample with Au and Ag single atoms co-existing on the surface of an ultra-thin carbon film on a Cu grid (Beijing XXBR Technology Co., Ltd), and imaged them with HAADF (Fig. R2a, b). From the contrast intensity analysis results shown in Fig. R2c,d, it can be seen that the Au single atoms have higher contrast intensities than the Ag, and their contrast intensities are both obviously higher than that of carbon.

Fig. R2 Contrast intensity comparison of Au and Ag single atoms co-existing on an ultra-thin carbon film. **a** HAADF image showing Au and Ag single atoms and clusters. They were produced using the

magnetron sputtering technique. Because the single atoms are all located on the surface of the ultra-thin carbon film, we can consider that in a small localized area, the single atoms are approximate all at a same plane. **b** Enlarged image of the single atoms in the box in **a**. It can be clearly seen that the contrasts of Au and Ag single atoms are different. **c** 3D contrast intensity analysis of **b**. **d** Contrast intensity profile in the dashed box in **b**, where the contrast of Au single atoms is obvious higher than that of Ag.

Based on the above contrast intensity analysis results of Au and Ag single atoms, we analyzed the contrast information of single atoms in thin amorphous carbon generated during the methane pyrolysis process, to further distinguish single atoms species. The results show that the contrast intensity of single atoms in amorphous carbon is relative uniform, and there is almost no difference in their contrast profile (Fig. R3b). And no Ag signals were detected in the amorphous carbon according to EDS results (Supplementary Figs. 4 and 11). Therefore, it can be determined that widely distributed and uniform Au single atoms have formed during methane pyrolysis.

Fig. R3 Contrast intensity analysis of single atoms in thin amorphous carbon generated through methane pyrolysis. **a** HAADF image of Au single atoms in ultrathin amorphous carbon. **b** The comparison chart of the single atoms contrast intensity randomly selected in **a**. It can be seen from **b** that there is no significant difference in the contrast of those single atoms. Thus, it can be speculated that widely distributed and uniform Au single atoms have formed during methane pyrolysis.

According to your friendly advice, we modified the description regarding the possible Ag content in amorphous carbon, as shown below:

"Importantly, HAADF imaging shows that a high density of heavy metal single atoms appear in the amorphous carbon (Fig. 1f and Supplementary Figs. 3–8 and 14) considering C is substantially lighter than Au or Ag^{4,8,21-26,28,38-40}. While no Ag signals were detected within the amorphous carbon (Supplementary Figs. 4 and 11), possibly due to its very low content in the original NPG sample, the existence of Ag single atoms is possible. Contrast intensity analysis in Supplementary Figs. 5–8 show the overwhelming majority of bright spots are Au single atoms, and a few darker bright spots may be Ag single atoms (Fig. 1f and Supplementary Figs. 3–6, 8, 14), especially considering the contrast intensity is also related to its distance from the focal plane (Supplementary Fig. 6)."

The above results have been added into the revised manuscript. They are presented on Pages 6 and 16 of the main text file and Pages 9, 10 and 11 of the Supplementary Information (SI) file, of which Figs. R2 and R3 are added as Supplementary Figs. 7 and 8.

Comment 3:

The intensity of the individual atom images and their clarity/width will depend on their height in the sample (i.e., whether they are on the top surface, in the interior, or on the bottom surface of the C). Were systematic through focal HAADF series taken to investigate the depth distribution of the atomic species? How does the visibility of the white patchy areas change with defocus?

Response 3:

Thank you very much for your constructive comments. According to your suggestions, we changed the focal plane and took HAADF images at different depth, which are shown in Fig. R4. Indeed, the contrast intensity analyses of the single atoms with different focus conditions show that their intensities are highest when they are near the focal plane; whereas, they become comparably lower under defocus conditions. Comparing with Fig. R4a and b, some singles atoms are bright at focus 0 nm and less bright at focus 2 nm (as indicated by circles 1–3 and Fig. R4c). And other single atoms are inverse (as indicated by circles 4–6 and Fig. R4d). Therefore, it can be speculated that single atoms (Au or possible Ag) are distributed in the whole amorphous carbon generated from the pyrolysis of methane.

Fig. R4 Contrast intensity analysis of single atoms with different focus in amorphous carbon generated through methane pyrolysis. **a,b** HAADF image of the Au single atoms in amorphous carbon with focus on 0 nm and 2 nm (relative height), respectively. **c,d** The comparison of the

same single atoms contrast profile in **a** and **b**. The contrast intensity of single atoms are sharper when they are near the focal plane. Whereas, their contrast intensities become lower under defocus conditions. The blue circles indicate the brighter single atoms, and the red circles indicates the lower bright single atoms in **a** and **b**, respectively.

It should be pointed out that during the process of taking HAADF images, the position of some single atoms may change (Fig. R4a, b). The sample area shown in Fig. R5 represents a relatively stable region, which provides us more accurate contrast intensity information for individual atoms. Again, single atoms and white patchy areas are sharper when they are in focus plane, while their intensities become weaker upon focus change.

Fig. R5 Contrast intensity analysis of stable single atoms and white patchy areas under different focus conditions. **a–c** HAADF image of the same Au single atoms and white patchy areas in amorphous carbon with focus on -5 nm, 0 nm and 8 nm (relative height), respectively. **d** The compared contrast profile of the same single atoms and white patchy areas in **a**, **b** and **c**, as indicated by circles 1 and boxes 2, respectively.

The above results have been added into the revised manuscript. They are presented on Page 6 of the main text file and Pages 6, 7 and 8 of the SI file, of which Figs. R4 and R5 are added as Supplementary Figs. 5 and 6.

Comment 4:

I am not entirely convinced the 'particles' shown in movies 4 and 5 are actually particles (- they look more like thin Au rafts).

Response 4:

Thank you very much for your constructive comments. During the methane pyrolysis process, the NPG catalyst undergoes dramatic structure evolution, featuring the formation of various Au nano- and subnano-structures as mentioned above. While Au rafts were indeed observed as discussed in our **Response 1** (Fig. R1a), they adopt quite different structures as compared to Au nanoparticles, which possess quite distinct crystal structures. For instance, the Au particles in Supplementary movies 5 and 4 clearly show the well-defined crystalline structure (Fig. R6a, b), and their lattice fringe spacing values were measured to be 2.06 Å and 2.05 Å, consistent with the standard value of Au{200}, 2.04 Å. The analysis of the crystal structures of the Au particles in Fig. R6c also gives the lattice fringe spacings and their angle value of 2.33 Å, 2.33 Å, and 70.8°, consistent with the standard value of Au, 2.35 Å, 2.35 Å, and 70.5°. In contrast, the Au raft in Fig. R1a (Fig. R6d) has the fringe spacings and their angles value of 2.39 Å, 2.37 Å, and 60.5°, obviously different from the standard values of Au. Therefore, we consider that the structures present in Supplementary movies 4 and 5 should be Au nanoparticles, not thin Au rafts.

Fig. R6 Crystal structures of Au nanoparticles and Au raft. The images in **a**, **b**, and **c** are Supplementary Fig. 12b, Fig. 3e, and Supplementary Fig. 11b, respectively. Supplementary Fig. 12b and Fig. 3e were taken from Supplementary Movies 5 and 4, respectively. **d** Microstructure of the Au raft in Fig. R1.

The above results have been added into the revised manuscript. They are presented on Page 11 of the main text file.

Comment 5:

I am also unconvinced that the particles in movies 4 and 5 are 'disintegrating'. They could just be re-orienting as the C film grows and moves. In fact, in Movie 5 it looks like the 200-type 'raft' like fringes disappear then later re-appear in approximately the same location as 111-type fringes.

Response 5:

Thank you very much for your constructive comments. Your suggestion about "They could just be re-orienting as the C film grows and moves" makes sense. The whole system, containing amorphous carbon, single atoms, clusters and particles, is in a highly dynamic process (Supplementary movies 4 and 5). Thus, the re-orienting of the Au particles might occur during the reaction process. Therefore, we re-checked the process and recorded a new movie (Supplementary Movie 6), some of whose moments are shown in Fig. R7a-f. The different orientations in Figs. R7a and R7d prove that re-orienting indeed occurred on the particle. But, the size reduction of the particle from 8.1 nm in Fig. R7a to 3.2 nm in Fig. R7e is rather large, and at the same moments the particle projection kept the nearly round shape. In consideration of 3D geometry, it is quite unlikely that sole re-orienting may cause such large size reduction while the nearly round projection shape maintains. Thus, it can be concluded that re-orienting and disintegrating simultaneously occurred on the particle.

Fig. R7 Structure evolution of an Au particle during the reaction process. This Au particle initially had the diameter of 8.1 nm (a), then continuously overturned and disintegrated (b-e), and finally disappeared (f). The insets in a and d are the enlarge images of the Au particle. Please see the detailed analysis in the text.

The above results and analyses have been added into the revised manuscript. They are presented on Page 11 of the main text file and Page 18 of the Supplementary Information (SI) file, of which Fig. R7 is added as Supplementary Fig. 13.

Comment 6:

Much of what is occurring at the Au/C interface and on the Au 'clean' surfaces is obscured by the strong delocalization effects (i.e., fringes extending beyond the edge of the gold) due to the FEG electron source. The authors should comment on this.

Response 6:

Thank you very much for your constructive comments. We agree that the delocalization effects cause lattice images to extend beyond the edges of the Au crystals and therefore mask some details that are occurring on the surfaces/interfaces.

In this article, we mainly focus on the dramatic dynamic changes of NPG structure during the catalytic methane pyrolysis. For example, Fig. 2 shows that at the Au/C interfaces, the surfaces of two Au ligaments lost the thicknesses of 2.6 nm (Fig. 2a–c) and 6.1 nm (Fig. 2d–f) within 2 second. In contrast, the fringe image part extending beyond the Au edge in Fig. 2a–c was only ~1 nm thick, and the one in Fig. 2d–f was almost invisible. Therefore, such dramatic and obvious loss of the Au surfaces at the Au/C interfaces cannot be obscured by the delocalization effects. The same phenomenon also occurred in Fig. 3d–f (namely Supplementary Movie 4) and Supplementary Movie 5.

Indeed, for too small evolutions of structures, the delocalization effects can cover up the changes. To address this issue, a TEM instrument with an image corrector (FEI Themis Z) was used. Its image corrector can eliminate the delocalization effects. As shown in Fig. R8, the Au ligament surface indicated by the white arrow is located at the appropriate focal plane, and thus its image does not have the delocalization effect. In contrast, the surface indicated by the black arrow is located at a different height, causing that its image has the delocalization effect. Further, Fig. R8 shows that the ligament surfaces are truly clean, and the 5-minute irradiation of the electron beam at 200 keV did not induce obvious structural changes on the surfaces, that is, the surfaces are stable. Therefore, we confirm that in the present research, it was the catalyzed pyrolysis of methane, rather than the electron beam irradiation, that induced the surface structure changes of NPG.

Fig. R8 HRTEM images of NPG surfaces before (a) and after (b) 5-minute continuous irradiation at 200 keV. These images were taken by FEI Themis Z with an image corrector. The delocalization effect does not and does exist on the ligament surface images indicated by the white and the black

arrows, respectively.

The above results and discussion have been added into the revised manuscript. They are presented on Pages 5, 14 and 17 of the main text file and Pages 29 and 30 of the SI file, of which Fig. R8 is added as Supplementary Fig. 23. The old version of Fig. 1c is replaced by Fig. R8a.

Comment 7:

Even with 1.4 ± 0.4 wt% Ag present (as measured by EDS), at least 1 in 100 of the dispersed atoms will still be Ag. It is also qualitatively noted that the catalytic activity decreases with decreasing Ag content. Does it drop linearly? The fact that the activity measurably changes with Ag content in this way suggests that they are indeed quite active species. If the Ag atoms are much more active compared to the Au, then it cannot be simply claimed that the behavior is dominated by the Au, and that the Ag contribution is insignificant.

Response 7:

Thank you very much for this constructive comment. In previous versions of this manuscript we were emphasizing that the very low Ag content should not be the primary reason for the drastic structural change of NPG, while one reviewer was more interested in the possible positive contribution of residual Ag for methane pyrolysis. It is absolutely reasonable that "If the Ag atoms are much more active compared to the Au, then it cannot be simply claimed that the behavior is dominated by the Au", and your comments gave us great inspiration.

According to your opinions, we first conducted more detailed literature search on the catalytic ability of Ag and Au toward methane pyrolysis. A series of transition metal clusters have been studied (*Au et al. J. Catal. 1999, 185, 12*; *Liao et al. J. Mol. Catal. A Chem. 1998, 136, 185*) for their abilities to catalyze the pyrolysis of methane, and the results are shown in Table R1. This table indicates that the overall ability of Ag to activate the C-H bonds of methane is **actually lower than** that of Au (Ag is slightly better than Au only in activating the fourth C-H bond of methane). Meanwhile, the values of the activation energy needed for the activation of the first C-H bond of CH₄ on metal-zeolite catalysts (metal: Ag or Au) also show that Ag and Au have similar catalytic activities (*Kurnaz et al. Micropor. Mesopor. Mat. 2011, 138, 68*). Considering the possible differences in actual structures of the catalysts (Au, Ag, AuAg alloys, nanoparticles or NPG with different size), it is thus reasonable to propose that the ability of Ag to catalyze the methane pyrolysis would **not be substantially stronger** than that of Au. This is the reason for us to consider that in the case of extremely low Ag content, the drastic dynamic structural change behavior of the NPG catalyst should be dominated by Au.

Table R1 Calculated Activation Energies (eV)[§] for C-H bond cleavage

Elementary reaction	Ag ₁₀	Au ₁₀
CH ₄ *→CH ₃ *+H*	1.54	1.42
CH ₃ *→CH ₂ *+H*	1.60	1.45
CH ₂ *→CH*+H*	1.61	1.17
CH*→C*+H*	1.90	1.95

*: Surface of the catalyst

[§]: Au et al. *Journal of Catalysis* 185: 12-22 1999; Liao et al. *Journal of Molecular Catalysis A: Chemical* 136: 185-194 1998.

In our experiment, we found that when the Ag content is increased to about 10%, the catalytic ability of NPG towards the methane pyrolysis <200 °C becomes active, while this ability is always active above 300 °C (Supplementary Fig. 26). This indeed indicates that substantial Ag incorporation may cause a new reaction mechanism for lower-temperature methane pyrolysis. Considering that in our present research, the reaction temperature was set at 346 °C, and in particular previous studies in this field have revealed significant structural dependence of NPG catalysts for different reactions (ref.³⁴⁻³⁶) (*Fujita et al. Nat. Mater. 2012, 11, 775*; *Liu et al. Adv. Mater. 2016, 28, 1753*; *Zhang et al. Catal. Sci. Technol. 2013, 3, 2862*), at this stage, we cannot conclude if there is a linear relationship between the Ag content and its catalytic activity for methane pyrolysis at different temperatures.

We thank you very much for your constructive comments again, and the above discussion and details have been added into the revised manuscript. They are presented on Pages 14, 15, 24 and 25 of the main text file and Page 39 of the SI file, of which Table R1 is added as Supplementary Table 1. The paper of *Au et al. J. Catal. 1999, 185, 12* is cited as ref 46; *Liao et al. J. Mol. Catal. A Chem. 1998, 136, 185* is cited as ref 47. *Kurnaz et al. Micropor. Mesopor. Mat. 2011, 138, 68* is cited as ref 48.

Comment 8:

Some of the terminology used by the authors (e.g violently changing, massive) while being very evocative is not very scientific.

Response 8:

Thank you very much. We have changed these descriptions accordingly in the revised manuscript.

REVIEWERS' COMMENTS:

Reviewer #4 (Remarks to the Author):

The authors have satisfactorily addressed all my questions and concerns and in doing so have further strengthened the manuscript. I now consider the paper suitable for publication in Nature Communications.